# DUAL PERSONALIZATION ON FEDERATED RECOMMENDATION

## ABSTRACT

Federated recommendation is a new Internet service architecture that aims to provide privacy-preserving recommendation services in federated settings. Existing solutions are used to combine distributed recommendation algorithms and privacy-preserving mechanisms. Thus it inherently takes the form of heavyweight models at the server and hinders the deployment of on-device intelligent models to end-users. This paper proposes a novel Personalized Federated Recommendation (PFedRec) framework to learn many user-specific lightweight models to be deployed on smart devices rather than a heavyweight model on a server. Moreover, we propose a new dual personalization mechanism to effectively learn fine-grained personalization on both users and items. The overall learning process is formulated into a unified federated optimization framework. Specifically, unlike previous methods that share exactly the same item embeddings across users in a federated system, dual personalization allows mild finetuning of item embeddings for each user to generate user-specific views for item representations which can be integrated into existing federated recommendation methods to gain improvements immediately. Experiments on multiple benchmark datasets have demonstrated the effectiveness of PFedRec and the dual personalization mechanism. Moreover, we provide visualizations and in-depth analysis of the personalization techniques in item embedding, which shed novel insights on the design of RecSys in federated settings.

## 1 INTRODUCTION

Federated recommendation is a new service architecture for Internet applications, and it aims to provide personalized recommendation service while preserving user privacy in the federated settings. Existing federated recommendation systems (Ammad-Ud-Din et al., 2019; Chai et al., 2020; Muhammad et al., 2020; Perifanis & Efraimidis, 2022; Singhal et al., 2021) are usually to be an adaptation of distributed recommendation algorithms by embodying the data locality in federated setting and adding privacy-preserving algorithms with guaranteed protection. However, these implementations of federated recommendations still inherit the traditional service architecture, which is to deploy large-scale models at servers. Thus it is impractical and inconsistent with the newly raised on-device service architecture, which is to deploy a lightweight model on the device to provide service independently without frequently communicating with the server. Given the challenge of implementing data locality on devices in federated settings, the personalization mechanism needs to be reconsidered to better capture fine-grained personalization for end-users.

Personalization is the core component of implementing federated recommendation systems. Inherited from conventional recommendation algorithms, existing federated recommendation frameworks are usually composed of three modules: user embedding to preserve the user's profile, item embedding to maintain proximity relationships among items, and the score function to predict the user's preference or rating for a given item. They usually preserve user-specific personalization in the user embedding module while sharing consensus on item embeddings and score functions.

This paper proposes a new **dual personalization** mechanism designed to capture fine-grained two-fold personal preferences for users in the federated recommendation system. Inspired by human beings' decision logic, we believe all modules in the recommendation framework should be used to preserve part of personalization rather than use user embedding only. For example, the score

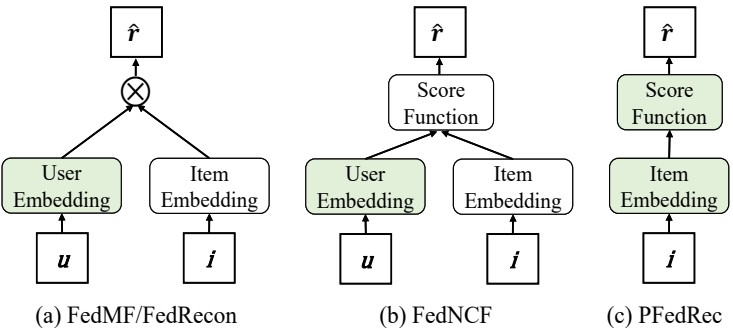

Figure 1: *Different frameworks for the personalized federated recommendation. The* *green block* *represents a* **personalized module**, *which indicates the part of model is to preserve user preference. Our proposed model will preserve dual personalization on two modules.*

function is to mimic the user's personal decision logic that is natural to be diverse across clients. Furthermore, given an itemset, different people may have a different view to measure their proximity relationships. Therefore, personalized item embedding could be essential to capture people's personal preferences further.

To implement the aforementioned ideas in federated settings, we propose a new federated recommendation framework to implement fine-grained personalization on multiple modules which are illustrated in Figure 1 (c). First, we use a personalized score function to capture users' preferences, and it could be implemented using a multi-layer neural networks. Second, we remove the user embedding from the framework because the current neural-based personalized score function has enough representation capability to preserve the information of user embeddings. Third, we implement a light fine-tuning to learn personalized item embeddings [1] in federated settings. This proposed decentralized intelligence architecture is a natural simulation of human beings' decision-making that each person has a relatively independent mind to make decisions.

The learning procedure is also carefully tailored in a federated setting. A personalized score function will be learned using its own data on the device, and then it won't be sent to the server for model aggregation that usually generates a general view for all devices. Moreover, the personalized item embedding will be implemented through light fine-tuning in a federated learning framework, thus it can leverage both the general view from server and the personalized view from its own data.

In summary, we propose a novel federated recommendation framework that integrates both the personalized score function and personalized item embedding via light finetuning from the shared item embedding. **Our key contributions** are summarized as follows.

- We propose a novel federated recommendation framework which is more naturally consistent with layer-wise neural architecture which can better fit federated learning.

- We design a novel dual personalization to capture user preferences using a personalized score function and to fine-grained personalization on item embeddings. It can be integrated with other baselines to improve their performances.

- We formulate the proposed federated recommendation learning problem into a unified federated optimization framework with the meta-learning objective.

- Our method can significantly outperform existing federated recommendation baselines.

## 2 RELATED WORK

### 2.1 PERSONALIZED FEDERATED LEARNING

**Federated Learning (FL)** is a new machine learning paradigm that a server orchestrates a large number of clients to train a model without accessing their data (Kairouz et al., 2021; Li et al., 2020; Bonawitz et al., 2019; Yang et al., 2019). The vanilla federated learning method, FedAvg (McMahan et al., 2017), is to learn a robust model at the server while embodying data locality for each device

---

[1]When the items are one-hot encoding vectors, we can simply equivalent use item embedding network and item embedding representations.

with non-IID data. **Personalized Federated Learning (PFL)** is to learn a personalized model for each device to tackle the non-IID challenge. A simple PFL method could train a global model with FedAvg, then conduct a few steps of finetuning on each client (Cheng et al., 2021). In this framework, knowledge sharing is model aggregation, and model personalization is local finetuning. Per-FedAvg (Fallah et al., 2020) adds finetuning as a regularization term to the objective learning function of the global model. Ditto (Li et al., 2021) proposes a bi-level optimization framework for PFL while constraining the distance between the local and global models. Investigations by (Shamsian et al., 2021; Chen et al., 2018) aim to train a global hyper-network or meta-learner instead of a global model before sending it to clients for local optimization. SCAFFOLD (Karimireddy et al., 2020) proposes to learn personalized control variate that corrects the local model accordingly. FedRecon (Singhal et al., 2021) is a meta-learning-based method that preserves a local model for each client and trains a global model collaboratively with FedAvg. Layer-wise personalization (Arivazhagan et al., 2019; Liang et al., 2020) is also a simple and effective technique in PFL.

## 2.2 FEDERATED RECOMMENDATION SYSTEMS

**Federated recommendation** has attracted much attention recently due to the rising concern about privacy. Some recent works focus on using the interaction matrix only. FCF (Ammad-Ud-Din et al., 2019) is the first FL-based collaborative filtering method, which employs the stochastic gradient approach to update the local model, and FedAvg is adopted to update the global model. Improving on user privacy protection, Chai *et al.* present FedMF (Chai et al., 2020), which adapts distributed matrix factorization to FL setting and introduces the homomorphic encryption technique on gradients before uploading to the server. MetaMF (Lin et al., 2020b) is a federated meta-learning framework where a meta-network is adopted to generate the rating prediction model and private item embedding. (Wu et al., 2021) presents FedGNN where each user maintains a GNN model to incorporate high-order user-item information. However, the server in both methods preserves all the recommendation model parameters which can be used to infer the user's interaction information, resulting in the risk of user privacy leakage. FedNCF (Perifanis & Efraimidis, 2022) adapts Neural Collaborative Filtering (NCF) (He et al., 2017) to the federated setting which introduces neural network to learn user-item interaction function to enhance the model learning ability. **Federated recommendation using rich information** considers multiple data sources or modalities in modeling. FedFast (Muhammad et al., 2020) extends FedAvg (McMahan et al., 2017) with an active aggregation method to facilitate the convergence. Efficient-FedRec (Yi et al., 2021) decomposes the model into a large news model on the server and a light user model on the client, and reduces the computation and communication cost for users. Both works rely on more data sources, such as user features or news attributes rather than an interaction matrix. (Lin et al., 2020a; Du et al., 2021; Yang et al., 2021; Minto et al., 2021; Lin et al., 2021) are endeavors that focus on enhancing privacy of FedRec. There are also attempts for other applications in FedRec, such as federated attack (Wu et al., 2022b; Zhang et al., 2022), social recommendation (Liu et al., 2022b), Click-Through Rate (CTR) prediction (Wu et al., 2022a), fair recommendation (Liu et al., 2022a) and payload optimization (Khan et al., 2021).

## 3 PROBLEM FORMULATION

**Federated Learning** is to learn a global model parameterized by $\theta$ to serve all clients whose data are private. The optimal solution should minimize the accumulated loss of all clients. That is,

$$\min_{\theta} \sum_{i=1}^{N} \alpha_i L_i(\theta) \tag{1}$$

where $L_i(\theta)$ is the supervised loss on the $i$-th client with dataset $D_i$, and all clients share the global parameter $\theta$. The $\alpha_i$ is a weight for the loss of the $i$-th client. For example, the conventional federated learning algorithm, FedAvg (McMahan et al., 2017), defines $\alpha_i$ as the fraction of the size of the client's training data, *i.e.,* $\alpha_i := |D_i| / \sum_{j=1}^{N} |D_j|$. Once the global model is trained, it can be used for prediction tasks on all clients.

**Personalized Federated Learning** simultaneously leverages common knowledge among clients and learns a personalized model for each client. The learning objective is usually formulated as

$$\min_{\theta, \{\theta_i\}_{i=1}^{N}} \sum_{i=1}^{N} \alpha_i L_i(\theta, \theta_i) \tag{2}$$

where each client has a unique personalized parameter $\theta_i$, and $\theta$ is the global parameter as mentioned in Eq. 1. For example, Fallah et al. (2020) leverage $\theta$ as initialization of $\theta_i$, *i.e.*, $\theta_i := \theta - \nabla l_i(\theta)$, where $l_i(\theta)$ is the loss of a vanilla model on the $i$-th client. The $L_i(\theta, \theta_i)$ is then formulated as

$$L_i(\theta, \theta_i) := l_i(\theta - \nabla l_i(\theta)) \tag{3}$$

**Recommendation on Neural Networks** This work focuses on a fundamental scenario where recommendation only relies on user-item interaction matrix. The recommendation task is then fulfilled by the Neural Collaborative Filtering (NCF) model (He et al., 2017), which consists of three parts: a score function $S$, a user embedding module $\mathcal{E}$ and an item embedding module $E$. We denote these modules' parameters as $\theta := (\theta^s, \theta^u, \theta^m)$ and formulate the learning objective in Eq. 4

$$\min_\theta L(\theta; r, \hat{r}) := \min_\theta L(\theta; r, S(\mathcal{E}(e^u), E(e^m))) \tag{4}$$

where $e^u$ and $e^m$ are one-hot encoding representing users and items. $r$ is a user's rate to the given item and $\hat{r}$ is a prediction from the score function $S(\mathcal{E}(e^u), E(e^m))$. $L$ is the loss evaluating prediction performance. It could be a ***point-wise loss*** as used in (Wang et al., 2016; He et al., 2017), or a ***pair-wise loss*** as in (Rendle et al., 2012; Wang et al., 2019). It is worth noting that conventional Matrix Factorization (MF) methods could be viewed as a special case of the NCF (He et al., 2017), *i.e.,* the conventional MF is a model where the score function $S$ is simplified as the multiplication operator without learnable parameters, and the embedding of user/item is obtained by the decomposition of the user-item interaction matrix.

# 4 METHODOLOGY

In this section, we propose a novel **P**ersonalized **Fed**erated **Rec**ommendation (PFedRec) framework, which aims to simultaneously learn many user-specific NCF models deployed on end devices.

## 4.1 OBJECTIVE FUNCTION

**Federated Learning Objective** We regard each user as a client under FL settings. The on-device recommendation task is then depicted as a PFL problem. Particularly, the item embedding module $E_i$ is assigned to be a global component which learns common item information and the score function $S_i$ is maintained locally to learn personalized decision logic. To further capture the difference between users and achieve a preference-preserving item embedding, we devise a bi-level optimization objective,

$$\min_{\theta^m, \{\theta_i\}_{i=1}^N} \sum_{i=1}^N \alpha_i L_i(\theta_i; r, \hat{r}) \tag{5}$$
$$s.t. \qquad \theta_i := (\theta^m - \nabla_{\theta^m} L_i, \theta_i^s)$$

where $\theta_i := (\theta_i^m, \theta_i^s)$ is the personalized parameter for $E_i$ and $S_i$, and $L_i$ will be evaluated on the $i$-th client local data $D_i$. Under this framework, PFedRec first tunes $E$ into a personalized item embedding module $E_i$, and then learns a lightweight local score function $S_i$ to make personalized predictions. Different from the conventional recommendation algorithms, the user embedding module $\mathcal{E}$ is depreciated since the personalization procedure on a client will automatically capture the client's preference. There is no use to learn extra embeddings to describe clients.

**Loss for Recommendation** Equipped with the item embedding module and score function, we formulate the prediction of $j$-th item by $i$-th user's recommendation model as,

$$\hat{r}_{ij} = S_i(E_i(e^j)) \tag{6}$$

Particularly, we discuss the typical recommendation task with implicit feedback, that is, $r_{ij} = 1$ if $i$-th user interacted with $j$-th item; otherwise $r_{ij} = 0$. With the binary-value nature of implicit feedback, we define the loss function of $i$-th user as the *binary cross-entropy loss*,

$$L_i(\theta_i; r, \hat{r}) = -\sum_{(i,j) \in D_i} \log \hat{r}_{ij} - \sum_{(i,j') \in D_i^-} \log(1 - \hat{r}_{ij'}) \tag{7}$$

where $D_i^-$ is the negative instances set of user. Notably, other loss functions can also be used, and here we choose the binary cross-entropy loss to simplify the description. Particularly, to construct $D_i^-$ efficiently, we first count all the uninteracted items set as,

$$\mathcal{I}_i^- = \mathcal{I} \backslash \mathcal{I}_i \qquad (8)$$

where $\mathcal{I}$ denotes the full item list and $\mathcal{I}_i$ is the interacted item set of $i$-th user. Then, we uniformly sample negative instances from $\mathcal{I}_i^-$ by setting the sampling ratio according to the number of observed interactions and obtain $D_i^-$.

### 4.2 Dual Personalization

We have implemented a dual personalization mechanism to enable the proposed framework can preserve fine-grained personalization for both user and item.

**Using partial-based federated model aggregation to learn personalized user score function on each device.** Our proposed model is composed of a neural-based score function parameterized by $\theta^s$ and an item embedding module parameterized by $\theta^m$. The coordinator/server of federated system will iteratively aggregate model parameters or gradients collected from each participant/device. Due to the concern of personalization and privacy, we could implement a partial model aggregation strategy by keeping the score function as a private module on devices while sharing the item embedding to the server. Therefore, the server only aggregates the gradients or parameters $\theta^m$ from the item embedding network. The user's personalized score function network $\theta^s$ won't be sent to the server and thus won't be aggregated. More discussion about learning efficiency of personalized score function can be found in **Appenidx A.1**.

**Fine-tuning the item embedding module to generate personalized representations for items on each device.** According to Eq. 5, the learning objective of $\theta^m$ could be viewed as searching for a "good initialization" that could be fast adaptive to the learning task on different devices. It shares similar ideas with meta-learning based methods (Fallah et al., 2020) which have a local loss in Eq. 3. However, our proposed method takes a different optimization strategy we call *post-tuning*. Specifically, rather than directly tuning a global model on clients' local data, it first learns the local score function with the global item embedding network, and then replaces the global item embedding with personalized item embedding obtained by fine-tuning $\theta^m$. Details of the learning process is illustrated in Algorithm. 1 and extensive experiments show it will achieve superior performance to the vanilla meta-learning based methods. Furthermore, we discuss the effectiveness of one-step fine-tuning in **Appendix A.2**.

### 4.3 Algorithm

**Optimization** To solve the optimization problem as descripted in Sec. 4.1 - objective function, we conduct an alternative optimization algorithm to train the model. As illustrated in Algorithm 1, when client receives the item embedding network from server, it first replace its embedding with global one, and then updates the score function while keeping item embedding network fixed. Then the client updates the item embedding based on the updated personalized score function. Finally, the updated item embedding would be uploads to server for global update.

**Workflow**. The overall workflow of the algorithm could be summarized into several steps as follows. The server is responsible for updating shared parameters and organizing all clients to complete collaborative training. At the beginning of federated optimization, the server initializes the model parameters, which would be used as initial parameters for all client models. In each round, the server selects a random set of clients and distributes the global item embedding $\theta^m$ to them. When local training is over, the server collects the updated item embedding network from each client to perform global aggregation. We build on the simplified version of FedAvg, a direct average of locally uploaded item embedding network. The overall procedure is summarized in Algorithm 1.

## 5 Discussions

### 5.1 Privacy on Federated Recommendation

Privacy-preserving is an essential motivation to advance existing cloud-centric recommendation to client-centric recommendation service architecture. In general, the federated learning's decen-

---

**Algorithm 1** Dual Personalization for Federated Recommendation

---

**ServerExecute:**
1: Initialize item embedding $\theta^m \leftarrow \theta_0^m$ and score function $\theta^s \leftarrow \theta_0^s$
2: **for** each round $t = 1, 2, ...$ **do**                    ▷ Global communication rounds
3:     $S_t \leftarrow$ (select a set of size $n$ randomly from all $N$ clients)
4:     **for** each client index $i \in S_t$ **in parallel do**
5:         $\theta_i^m \leftarrow$ ClientUpdate$(i, \theta^m)$      ▷ Distribute global item embedding to client for update
6:     $\theta^m \leftarrow \frac{1}{n} \sum_{i=1}^{n} \theta_i^m$                ▷ Global aggregation over $n$ local item embedding network

**ClientUpdate:**
1: Initialize $\theta_i^m$ with $\theta^m$
2: Initialize $\theta_i^s$ with the latest updates
3: Retrieve user positive feedback $D_i$ according to user index $i$
4: Sample negative feedbcak $D_i^-$ from unobserved instances
5: $\mathcal{B} \leftarrow$ (split $D_i \cup D_i^-$ into batches of size $B$)
6: **for** $e$ from 1 to $E$ **do**                              ▷ Local training epochs
7:     **for** batch $b \in \mathcal{B}$ **do**
8:         Compute $L_i(\theta_i; r, \hat{r})$ with Eq. 7              ▷ Model loss of batch data $b$
9:         $\theta_i^s \leftarrow \theta_i^s - \eta \nabla_{\theta^s} L_i$                    ▷ Score funtion update
10:        Compute $L_i(\theta_i; r, \hat{r})$ with Eq. 7          ▷ Model loss with the updated $\theta_i^s$
11:        $\theta_i^m \leftarrow \theta_i^m - \eta' \nabla_{\theta^m} L_i$        ▷ post-tuning for personalized item embedding network
12: **Return** $\theta_i^m$ to server

---

tralized framework can embody data locality and information minization rules (GDPR) that could greatly mitigate the risk of privacy leakage (Kairouz et al., 2019). To provide service with privacy guranttee, the FL framework should be integrated with other privacy-preserving methods, such as Differential Privacy and secure communication. Our proposed framework derives the same decentralized framework from vaniall FL to preserve data locality. For example, to tackle the privacy leakage risk caused by sending item embedding network to the server, we could simply apply differential privacy to inject noise into the gradients so that the server cannot simply infer the updated items by watching the changes of gradients. More discussion can be found in **Appendix D.4**

### 5.2 A GENERAL FRAMEWORK FOR FEDERATED RECOMMENDATION

The proposed framework in Figure 1 (c) could be a general form of federated recommendation. Because our framework could be easily transformed to an equivalent form of other frameworks. For example, if we assign the score function as a one-layer neural network, PFedRec is equal to FedMF and FedRecon in Figure 1 (a). Moreover, if we change the personalized score function from full personalization to partial layer personalization, our method could be equivalent to FedNCF in Figure 1 (b) which has a shared score function across clients. Furthermore, our proposed framework's architecture could be naturally aligned to the classic neural network architecture, thus it has a bigger potential to achieve a better learning efficiency and is more flexible to extend. More discussion about communication efficiency and time complexity can be found in **Appendix B.1 and B.2**

## 6 EXPERIMENTS

### 6.1 EXPERIMENTAL SETUP

We evaluate the proposed PFedRec on four real-world datasets: MovieLens-100K, MovieLens-1M, Lastfm-2K and Amazon-Video. They are all widely used datasets in assessing recommendation models. Specifically, two MovieLens datasets were collected through the MovieLens website, containing movie ratings and each user has at least 20 ratings. Lastfm-2K is a music recommendation dataset, and each user maintains a list of her favorite artists and corresponding tags. Amazon-Video was collected from the Amazon site, containing product reviews and metadata information. We excluded users with less than 5 interactions in Lastfm-2K and Amazon-Video. The characteristics of datasets are shown in **Appendix C.1**. For dataset split, We follow the prevalent leave-one-out evaluation (He et al., 2017). We evaluate the model performance with Hit Ratio (HR) and Normalized Discounted Cumulative Gain (NDCG) metrics. Details can be referred to **Appendix C.2**.

## 6.2 BASELINES AND IMPLEMENTATION DETAILS

**Baselines** Our method is compared with baselines in both centralized and federated settings. Focusing on the performance improvement of the infrastructure of recommendation models that all others derive from, we select the general and fundamental baselines that conduct recommendations based on the interaction matrix.

- **Matrix Factorization (MF)** (Koren et al., 2009): This method is a typical recommendation algorithm. Particularly, it decomposes the rating matrix into two embeddings located in the same latent space to characterize user and item, respectively.
- **Neural Collaborative Filtering (NCF)** (He et al., 2017): This method models user-item interaction function with an MLP, and is one of the most representative neural recommendation models. Specifically, we apply the interaction function with a three-layers MLP for comparison, adopted in the original paper.
- **FedMF** (Chai et al., 2020): It is a federated version of MF and a typical FedRec method. It updates user embedding locally and uploads item gradients to the server for global update.
- **FedNCF** (Perifanis & Efraimidis, 2022): It is a federated version of NCF. Specifically, each user updates user embedding locally and uploads item embedding network and score function to the server for global update.
- **Federated Reconstruction (FedRecon)** (Singhal et al., 2021): It is a state-of-the-art PFL framework, and we test it under the matrix factorization scenario. Between every two rounds, this method does not inherit user embedding from the previous round but trains it from scratch.

**Implementation details** We randomly sample 4 negative samples for each positive sample following (He et al., 2017). For a fair comparison, we keep the same latent user (item) embedding size for all methods, *i.e.,* 32 and set other model details of baselines according to their original papers. For our method, we assign the score function with a one-layer MLP for simplification, which can be regarded as an enhanced FedMF with the dual personalization mechanism. We implement the methods based on the Pytorch framework and run all the experiments for 5 repetitions and report the average results. Parameter configuration can be found in **Appendix C.3**

## 6.3 COMPARISON ANALYSIS

We conduct experiments on four datasets for performance comparison.

| | Method | MovieLens-100K | | MovieLens-1M | | Lastfm-2K | | Amazon-Video | |
|---|---|---|---|---|---|---|---|---|---|
| | | HR@10 | NDCG@10 | HR@10 | NDCG@10 | HR@10 | NDCG@10 | HR@10 | NDCG@10 |
| Rec | NCF | 64.14 ± 0.98 | 37.91 ± 0.37 | 64.17 ± 0.99 | 37.85 ± 0.68 | 82.44 ± 0.42 | 67.43 ± 0.89 | 60.16 ± 0.43 | 38.97 ± 0.14 |
| | MF | 64.43 ± 1.02 | 38.95 ± 0.56 | 68.45 ± 0.34 | 41.37 ± 0.18 | 82.71 ± 0.54 | 71.04 ± 0.62 | 46.69 ± 0.65 | 29.83 ± 0.45 |
| FedRec | FedNCF | 60.62 ± 0.59 | 33.25 ± 1.35 | 60.54 ± 0.46 | 34.17 ± 0.40 | 81.55 ± 0.38 | 61.03 ± 0.63 | 57.77 ± 0.07 | 36.86 ± 0.06 |
| | FedRecon | 64.45 ± 0.81 | 37.78 ± 0.38 | 63.28 ± 0.15 | 36.59 ± 0.33 | 82.06 ± 0.38 | 67.58 ± 0.35 | 59.80 ± 0.14 | 38.87 ± 0.13 |
| | FedMF | 65.15 ± 1.16 | 39.38 ± 1.08 | 67.72 ± 0.14 | 40.90 ± 0.14 | 81.64 ± 0.48 | 69.36 ± 0.42 | 59.67 ± 0.19 | 38.55 ± 0.21 |
| | PFedRec (Ours) | **71.62 ± 0.83** | **43.44 ± 0.89** | **73.26 ± 0.20** | **44.36 ± 0.16** | **82.38 ± 0.92** | **73.19 ± 0.38** | **60.08 ± 0.08** | **39.12 ± 0.09** |

Table 1: *Performance of HR@10 and NDCG@10 on four datasets.* ***Rec*** *and* ***FedRec*** *represent centralized and federated methods, respectively. The results are the mean and standard deviation of five repeated trials.*

**Results & discussion** As shown in Table 1, *(1)* PFedRec obtains better performance than centralized methods in some cases. In centralized scenario, only user embedding is regarded as personalized component to learn user characteristics, and other components are totally shared among users. In comparison, our dual personalization mechanism considers two forms of personalization, which can further exploit user preferences. *(2)* PFedRec realizes outstanding advances on the two Movie-Lens datasets. These two contain more samples for training, which supports the personalization depiction. *(3)* PFedRec consistently achieves the best performance on all settings. In FedRec, the common item embedding helps transfer the shared information among users, which facilitates collaborative training of individual user models. However, different users present rather distinct preferences for items. Our dual personalization mechanism offers fine-grained personalization which fits the local data. It filters out the interference of redundant information and obtains better performance.

We also analyse spacial and computational efficiency of our proposed model and compare with baselines, including parameter volume, training epoch and running time. Specifically, our proposed

method achieves best performance with relatively low level of running time consumption. It shows that our dual personalization mechanism will not increase spatial and computational compexity. Detailed analysis could be found in **Appendix D.1**. Besides, we present the convergence curves of PFedRec and all baselines on all datasets in **Appendix D.2**. It shows PFedRec is the consistently the fastest one to converge, which emphasizes our method's advanced efficiency.

## 6.4 INTEGRATING BASELINES WITH OUR DUAL PERSONALIZATION MECHANISM

This paper proposes a lightweight dual personalization mechanism to enhance personalization handling, which can be easily transferred to nearly-all federated learning methods. We apply it to FedRec baselines to exhibit its efficacy.

| Method | MovieLens-100K | | MovieLens-1M | | Lastfm-2K | | Amazon-Video | |
|---|---|---|---|---|---|---|---|---|
| | HR@10 | NDCG@10 | HR@10 | NDCG@10 | HR@10 | NDCG@10 | HR@10 | NDCG@10 |
| **FedNCF** | $60.62 \pm 0.59$ | $33.25 \pm 1.35$ | $60.54 \pm 0.46$ | $34.17 \pm 0.40$ | $81.55 \pm 0.38$ | $61.03 \pm 0.63$ | $57.77 \pm 0.07$ | $36.86 \pm 0.06$ |
| **w/ DualPer** | $68.82 \pm 1.35$ | $39.33 \pm 0.85$ | $68.17 \pm 0.55$ | $39.56 \pm 0.29$ | $82.31 \pm 0.56$ | $71.64 \pm 0.43$ | $59.57 \pm 0.57$ | $38.73 \pm 0.62$ |
| **Improvement** | ↑ **13.53%** | ↑ **18.29%** | ↑ **12.60%** | ↑ **15.77%** | ↑ 0.93% | ↑ **17.38%** | ↑ 3.12% | ↑ **5.07%** |
| **FedRecon** | $64.45 \pm 0.81$ | $37.78 \pm 0.38$ | $63.28 \pm 0.15$ | $36.59 \pm 0.33$ | $82.06 \pm 0.38$ | $67.58 \pm 0.35$ | $59.80 \pm 0.14$ | $38.87 \pm 0.13$ |
| **w/ DualPer** | $70.20 \pm 0.90$ | $41.83 \pm 0.71$ | $68.89 \pm 0.26$ | $40.04 \pm 0.16$ | $83.51 \pm 0.23$ | $74.83 \pm 0.44$ | $60.23 \pm 0.16$ | $39.20 \pm 0.12$ |
| **Improvement** | ↑ **8.92%** | ↑ **10.72%** | ↑ **8.87%** | ↑ **9.43%** | ↑ 1.77% | ↑ **10.73%** | ↑ 0.72% | ↑ 0.85% |
| **FedMF** | $65.15 \pm 1.16$ | $39.38 \pm 1.08$ | $67.72 \pm 0.14$ | $40.90 \pm 0.14$ | $81.64 \pm 0.48$ | $69.36 \pm 0.42$ | $59.67 \pm 0.19$ | $38.55 \pm 0.21$ |
| **w/ DualPer** | $71.62 \pm 0.83$ | $43.44 \pm 0.89$ | $73.26 \pm 0.20$ | $44.36 \pm 0.16$ | $82.38 \pm 0.92$ | $73.19 \pm 0.38$ | $60.08 \pm 0.08$ | $39.12 \pm 0.09$ |
| **Improvement** | ↑ **9.93%** | ↑ **10.31%** | ↑ **8.18%** | ↑ **8.46%** | ↑ 0.91% | ↑ **5.52%** | ↑ 0.69% | ↑ 1.48% |

Table 2: *Performance improvement for integrating our dual personalization mechanism to baseline algorithms.* **Improvement** *denotes the performance gain beyond the baselines due to incorporating our dual personalization mechanism (**DualPer**). The results are the mean and standard deviation of five repeated trials, and the significant improvements (over 5%) are highlighted.*

**Results & discussion** According to Table 2, all the baselines are significantly improved by integrating our dual personalization mechanism since our mechanism enhances their modeling of user personalization. The highest HR and NDCG increases exist at FedNCF on MovieLens-100K, *i.e.,* 13.53% and 18.29%. The enhancement of FedNCF attains the most remarkable boost, which emphasizes the necessity of learning personalized item embedding for each user and the capacity of our dual personalization mechanism. Comparing with Lastfm-2K and Amazon-Video, the improvement of this mechanism is more evident on the two MovieLens datasets, almost around 10%, where each user has more samples locally. In summary, our proposed dual personalization mechanism can help the local model to learn more information when training personalized item embedding, which benefits the recommendation system prominently.

**Impact Factor Analysis** To further analysis the efficacy of PFedRec, we design comprehensive impact factor influence analysis, including latent embedding size, negative sample size and clients volume participationg in each round. In a nutshell, the best embedding size for all datasets is 32. Generally, the performance grows gradually as the negative samples increases. PFedRec is able to obtain the best performance regardless of different volume of client samples, while more clients facilitate the convergence. For more details please refer to **Appendix D.3**.

## 6.5 A CLOSE LOOK OF PERSONALIZATION IN PFEDREC

In our method, we learn the personalized item embedding network for each user based on the global item embedding to learn fine-grained personalization. To further verify and analyze the role of personalized item embedding, we conduct empirical experiments to answer the questions:

• *Q1: Why personalized item embeddings benefit recommendation more than the global one?*

• *Q2: How specific are the personalized item embeddings between users?*

**To answer *Q1***, We first discuss its straightforward insight, then we present visualization to demonstrate our claim. The recommendation system is supposed to provide user-specific recommendations by exploiting historical interactions. In the FedRec setting, item embedding is consistently considered to maintain the common characteristics among users, and its role in depicting user-specific preferences has been neglected. On the other hand, describing users with common item embedding leads in noisy information, which may incur unsuitable recommendations. Through personalizing item embedding, we enhance personalization modeling in federated learning methods, which depicts the user-specific preference. We will demonstrate how precise the personalized embedding describes user preference of our method compared with baselines.

We compare the item embedding learned by baselines with our method. Particularly, we select a user randomly from the MovieLens-100K dataset and visualize the embeddings by mapping them into a 2-D space through t-SNE (Maaten & Hinton, 2008). In this paper, we mainly focus on the implicit feedback recommendation, so each item is either a positive or negative sample of the user. As shown in Figure 2, the item embeddings of positive (blue) and negative (purple) samples are mixed together in baselines. However, they can be obviously divided into two clusters by PFedRec. We can easily conclude that our model learns which items the user prefers.

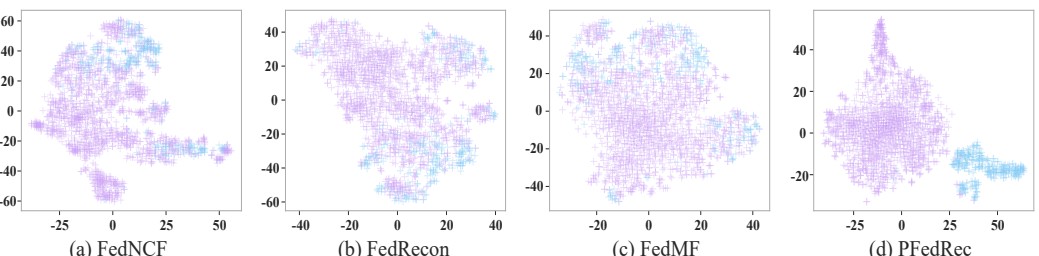

Figure 2: *TSNE visualization of item embeddings learnd by baselines and our method.*

**To answer *Q2***, we compare three usages of item embedding as follows:

– *Another random user*: Each client is assigned with item embedding from another random user, *i.e.,* every client runs with its score function and item embedding from another random user.

– *Global*: We assign each client with globally shared item embedding, *i.e.,* every client runs with its score function and global item embedding.

– *Own*: It follows our setting that every client runs with its own score function and item embedding.

Specifically, we first train PFedRec, then assign the learned item embeddings as the above three ways for inference. Experimental results are shown in Figure 3. Clients with their item embedding achieve the best performance, and clients with item embedding from others degrade significantly. Item embedding from another user contains a relatively low level of helpful information for inference, even less than the common characteristics in global item embedding. The personalized item embedding learned by PFedRec has been adapted to the client preference, and different clients achieve rather distinct item embeddings, which depict the user-specific information.

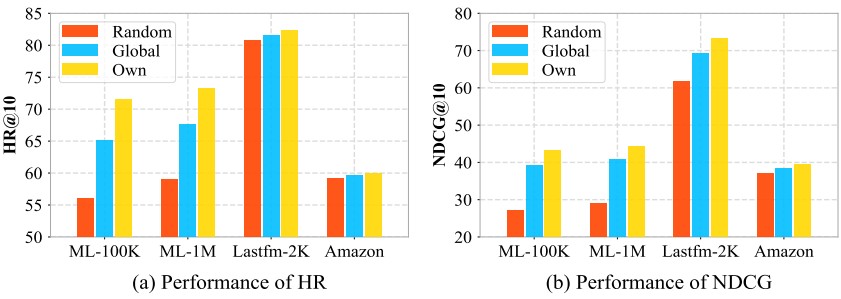

Figure 3: *Client inference using different item embeddings.*

# 7 CONCLUSION

This paper proposes a novel personalized federated recommendation framework to learn many on-device models simultaneously. We are the first to design the dual personalization mechanism that can learn fine-grained personalization on both users and items. This work could be a fundamental work to pave the way for implementing a new service architecture with better privacy preservation, fine-grained personalization, and on-device intelligence. Given the complex nature of modern recommendation applications, such as cold-start problems, dynamics, using auxiliary information, and processing multi-modality contents, our proposed framework is simple and flexible enough to be extended to tackle many new challenges. Moreover, the proposed dual personalization is a simple-but-effective mechanism to be easily integrated with existing federated recommendation systems.

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

## A  MORE DISCUSSION ABOUT DUAL PERSONALIZATION

### A.1  LEARNING EFFICIENCY OF PERSONALIZED SCORE FUNCTION

Due to the design of dual personalization, the user's critical information is decomposed into personalized item embeddings and a personalized score function. Therefore, in most scenarios, including inactive users, the score function module does not demand a large neural network. The simple and swift multi-layer neural network is capable of tackling most scenarios. For example, in our implementation, we use a one-layer MLP as the score function, which achieves significant performance improvement again baselines. Particularly, we conduct empirical experiments to analyze the model's effectiveness on inactive users and details can be found in D.6. Moreover, to further improve the learning efficiency, we could pre-train a neural score model for each user group considering their demographic features.

### A.2  EFFECTIVENESS OF ONE-STEP FINE-TUNING

The one-step gradient fine-tuning is necessary for our proposed PFedRec, and our design is supported by the theoretical analysis in Pillutla et al. (2022). A significant consideration is that the PFedRec alternately tunes one gradient step each for the global item embedding module and the client-specific score function. This learning process is consistent with the partial personalization scheme studied in Pillutla et al. (2022), where theoretical analysis shows that one gradient step is sufficient and the algorithm's convergence is guaranteed when the following assumptions hold: *(1)* The gradient of a local loss function is L-Lipschitz smooth. *(2)* The gradient of a local loss function has bounded variance. *(3)* the variance of the gradient of local loss functions on different clients is bounded. As the three assumptions are universal and hold for most functions, the conclusions from

Pillutla et al. (2022) also hold in our proposed PFedRec, i.e., the one-step fine-tuning is sufficient and guaranteed to converge.

# B KEY ISSUES DISCUSSION IN FEDERATED RECOMMENDATION

## B.1 COMMUNICATION EFFICIENCY

Due to the nature of federated learning, multiple rounds of parameter transfers are required between the server and the clients to complete the training process, and communication efficiency is an important consideration in federated learning modeling. For the FedRec model, there are three key factors that determine the communication cost, including the parameter volume $\mathcal{M}$ transmitted between server and clients, the number of sampled clients $\mathcal{S}$ in each round, and the total communication rounds $\mathcal{T}$. The overall communication cost can be formulated as,

$$\mathcal{C} = \mathcal{M} \cdot \mathcal{S} \cdot \mathcal{T} \tag{9}$$

Since $\mathcal{S}$ and $\mathcal{T}$ are constant among different models, the communication cost is positively correlated to $\mathcal{M}$, *i.e.,* $\mathcal{C} \propto \mathcal{M}$. Generally, item embedding is regarded as the shared component in FedRec research. For example, FedMF Chai et al. (2020) and FedRecon Singhal et al. (2021) only transmit item embedding whose $\mathcal{M} = |\mathcal{I}| \cdot d$, where $|\mathcal{I}|$ is the number of items and $d$ denotes the embedding size. Besides, there are also some works that take the score function or user embedding as common components transmitted between server and clients together with item embedding. FedNCF Perifanis & Efraimidis (2022) and MetaMF Lin et al. (2020b) transmit both item embedding and score function whose $\mathcal{M} = |\mathcal{I}| \cdot d + |\mathcal{F}|$, where $|\mathcal{F}|$ is the parameter volume of score function, such as a three-layer MLP. FedGNN Wu et al. (2021) also share user embedding whose $\mathcal{M} = |\mathcal{I}| \cdot d + |\mathcal{F}| + |\mathcal{U}| \cdot d$, where $|\mathcal{U}|$ is the number of users. In our method, we only share item embedding among users, which results in the minimal transmission parameter volume, *i.e.,* $\mathcal{M} = |\mathcal{I}| \cdot d$.

In our implementation, each client does not need to maintain the full item embedding table. Since each user only trains her local model with historical interactions and randomly sampled negative instances[2], *e.g.,* 5 times of historical interactions. As a result, the number of items processed by the user is much less than the complete item list, which further improves the communication efficiency of the proposed method. Take Lastfm-2K as an example, the size of the item set is 12,454, and the average size of historical interactions is 116, which is further less than transmitting the full item set. Moreover, in this paper, we mainly focus on the fundamental recommendation task that only relies on the interaction matrix, and it is necessary to instantiate the item embedding module with the item embedding table. When implementing our framework under the scenario where item attributes are available, we can replace the item embedding table with a lightweight neural network similar to the score function. Then the local model on each device can be more lightweight, which results in less communication cost. More empirical results about communication efficiency can be found in **Efficiency Comparison** D.1, **Convergence Analysis** D.2 and **Hyper-Parameter Study–Effect of client samples participating in each round** D.3.

## B.2 TIME COMPLEXITY

We analyze the time complexity of PFedRec. For clarity, we restate the notation that $|\mathcal{I}| \cdot d$ is the item embedding table, where $|\mathcal{I}|$ is the number of items and $d$ is the latent embedding size; $\theta^s$ is the score function, and we instantiate it with a single-layer MLP whose size is $d$; let $e$ denotes the local training epochs, $\mathcal{S}$ is the sampled clients in each round and $\mathcal{T}$ is the total communication rounds. Since $e$ is usually a small constant, *e.g.,* $e = 1$ in our implementation, then the time complexity of PFedRec is $\mathcal{O}(|\mathcal{I}|d^2\mathcal{S}\mathcal{T})$.

# C EXPERIMENTAL SETUP DETAILS

## C.1 MORE DETAILS OF THE DATASETS

---

[2]For practical implementation, the server could randomly sample ordinary instances rather than negative instances only, and then send it to the client to judge whether it is a negative instance. The server's item set size is usually much bigger than the number of interacted items by a user. Thus most randomly selected instances will be negative instances.

We filter the users whose historic interactions are less than 5 and the characteristics of the four datasets used in experiments are shown in Table 3. Since we focus on the implicit feedback recommendation scenario in this paper, we transform the ratings in each dataset into implicit data, that is, let 1 mark that the user has rated an item.

| Dataset | Interactions | Users | Items | Sparsity |
|---|---|---|---|---|
| MovieLens-100K | 100,000 | 943 | 1,682 | 93.70% |
| MovieLens-1M | 1,000,209 | 6,040 | 3,706 | 95.53% |
| Lastfm-2K | 185,650 | 1,600 | 12,454 | 99.07% |
| Amazon-Video | 63,836 | 8,072 | 11,830 | 99.93% |

Table 3: Datasets statistics.

## C.2 EVALUATION PROTOCOLS

For dataset split, we adopt prevalent leave-one-out evaluation, following He et al. (2017). For each user, we take her last interaction as the test data and remain other interactions for training. Additionally, we regard the latest raction in the training set as the validation data to find hyper-parameters. For the final evaluation, we follow the regular strategy Koren (2008); He et al. (2017) that randomly samples 99 unobserved items for each user, ranking the test item among the 100 items. We evaluate the ranked list with Hit Ratio (HR) and Normalized Discounted Cumulative Gain (NDCG) metrics. In particular, HR measures whether the test data is in the top-K list, while NDCG considers the test data's position in the list. The two evaluation metrics could be formulated as follows,

$$HR = \frac{1}{N} \sum_{u=1}^{N} hits(u) \tag{10}$$

where $N$ is the number of users and $hits(u) = 1$ indicates that the test data of user $u$ is in the top-K recommendation list, otherwise $hits(u) = 0$.

$$NDCG = \frac{1}{N} \sum_{u=1}^{N} \frac{\log 2}{\log(p_u + 1)} \tag{11}$$

where $p_u$ denotes the position of test data of user $u$ in the recommendation list and $p_u \rightarrow \infty$ when the test data is not in the top-K recommendation list. Here we set $K = 10$ for all experiments.

## C.3 PARAMETER CONFIGURATION

In the implementation, we set the latent embedding size as 32 and the batchsize is fixed as 256 for all baselines and our method. The total number of communication rounds is set to 100, and this value enables all methods to be trained to converge through experiments. For learning rate, we search it in $[0.0001; 0.0005; 0.001; 0.005; 0.01; 0.05; 0.1; 0.5]$ and the specific setting on four datasets are summarized in Table 4. For other empirical experiments of our method, we directly used the same parameters without researching.

| Method | ML-100K | ML-1M | Lastfm-2K | Amazon |
|---|---|---|---|---|
| NCF | 0.005 | 0.005 | 0.005 | 0.005 |
| MF | 0.0005 | 0.0005 | 0.001 | 0.001 |
| FedNCF | 0.5 | 0.5 | 0.5 | 0.5 |
| FedNCF w/ DualPer | 0.5 | 0.5 | 0.5 | 0.5 |
| FedRecon | 0.1 | 0.1 | 0.1 | 0.05 |
| FedRecon w/ DualPer | 0.1 | 0.1 | 0.1 | 0.1 |
| FedMF | 0.1 | 0.1 | 0.05 | 0.05 |
| PFedRec (Ours) | 0.1 | 0.1 | 0.05 | 0.01 |

Table 4: Learning rate configuration for all methods on four datasets. **w/ DualPer** indicates the model enhanced with our proposed dual personalization mechanism.

# D  MORE EXPERIMENTAL RESULTS

## D.1  EFFICIENCY COMPARISON

In federated learning, space and time efficiency are prominent factors for application. We compare the model's efficiency, including parameter volume, training epochs and running time, as shown in Table 5. According to the results, *(1)* FedNCF has the largest volumes of parameters. Parameters of recommendation systems consist of several parts, *i.e.,* user and item embedding and score function. The embedding size is the same for all methods in each dataset. For score function, FedNCF employs a three-layers MLP, which leads to much more parameters than one-dimensional embedding in other methods. *(2)* FedRecon takes the most training epochs to converge. The reconstruction mechanism in FedRecon demands retraining the local module from scratch in each round, which results in more training epochs. Taking MovieLens-100K as an example, our method converges in 61 training epochs, while FedRecon requires 465 training epochs, approaching 8 times ours. FedNCF takes the second longest training epochs and training time. On MovieLens-100K, the total training time of FedNCF is about 2 times as long as ours. *(3)* The space and time efficiency of FedMF is at the same level as our method. Our method enhances the personalization modeling and achieves the best performance without extra computational complexity. In our experiments, we found that just one local gradient descent step to learn personalized item embedding can yield advanced performance.

| Method | MovieLens-100K | | | MovieLens-1M | | | Lastfm-2K | | | Amazon-Video | | |
| --- | --- | --- | --- | --- | --- | --- | --- | --- | --- | --- | --- | --- |
| | parameters | epochs | time (s) | parameters | epochs | time (s) | parameters | epochs | time (s) | parameters | epochs | time (s) |
| FedNCF | 86,753 | 90 | 1,800 | 314,625 | 98 | 18,718 | 452,481 | 95 | 4,750 | 639,617 | 60 | 5,820 |
| w/ finetune | 86,753 | 90 | 1,876 | 314,625 | 98 | 19213 | 452,481 | 83 | 4,680 | 639,617 | 32 | 4,120 |
| FedRecon | 53,856 | 465 | 9,486 | 118,624 | 470 | 123,892 | 398,560 | 98 | 7,154 | 378,592 | 57 | 3,819 |
| w/ finetune | 53,856 | 444 | 9,380 | 118,624 | 476 | 125,230 | 398,560 | 87 | 6,740 | 378,592 | 91 | 4,760 |
| FedMF | 53,856 | 72 | 936 | 118,624 | 93 | 12,927 | 398,560 | 82 | 3,280 | 378,592 | 56 | 3,920 |
| PFedRec | 53,857 | 61 | 854 | 118,625 | 95 | 13,585 | 398,561 | 60 | 2,340 | 378,593 | 75 | 5,100 |

Table 5: Efficiency comparison results on four datasets, including model parameter volume, training epochs and running time.

## D.2  CONVERGENCE ANALYSIS

We show the convergence curves of the two evaluation metrics for our method and baselines on four datasets. As shown in Figure 4, our method achieves the fastest convergence on all datasets and metrics almost all the time, followed by FedMF and FedRecon. FedNCF has the slowest convergence speed because it has the most parameters and requires a longer number of iterations.

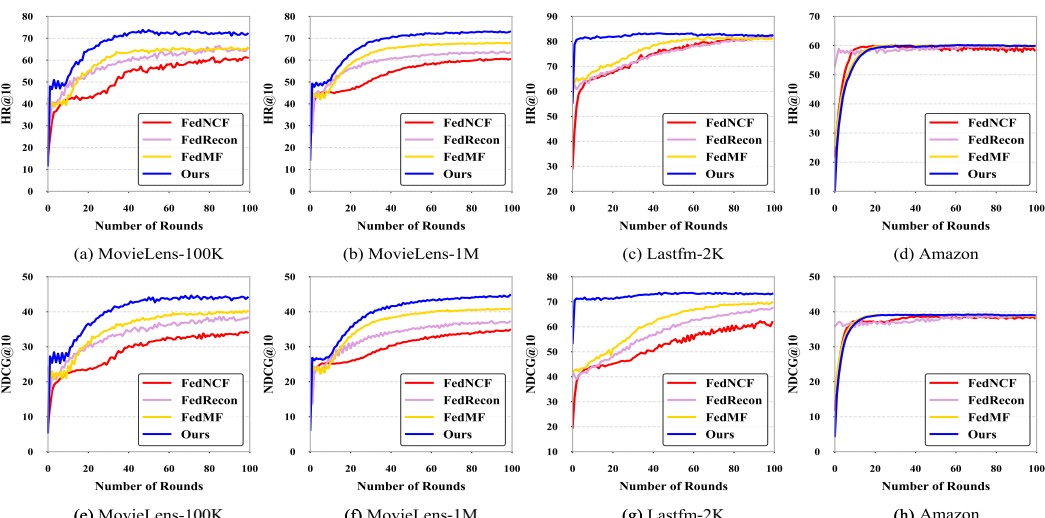

Figure 4: Model convergence comparison. The horizontal axis is the number of federated optimization rounds, and the vertical axis is the model performance, where (a)-(d) are HR@10 metric and (e)-(h) are NDCG@10 metric.

### D.3 HYPER-PARAMETERS STUDY

**Effect of latent embedding size** We tune the latent embedding size from $\{16, 32, 64, 128\}$. Accordingly, the architecture of the score function, *i.e.,* one-layer MLP, is $16 \rightarrow 1$, $32 \rightarrow 1$, $64 \rightarrow 1$ and $128 \rightarrow 1$, respectively. Experimental results are shown in Table 6. Almost all four datasets achieve the best results when the latent embedding size is 32. Due to the limited data of a single user, increasing the model parameters did not obtain further performance improvement. Generally, setting the dimension to 16 or 32 can achieve advanced performance. A small volume of parameters helps to build a lightweight on-device recommendation model.

| Dataset | Metrics | 16 | 32 | 64 | 128 |
|---------|---------|-----|-----|-----|-----|
| Movielens-100K | HR@10 | **72.81 $\pm$ 0.90** | 71.62 $\pm$ 0.83 | 71.64 $\pm$ 0.44 | 71.75 $\pm$ 0.80 |
| | NDCG@10 | 43.32 $\pm$ 0.43 | 43.44 $\pm$ 0.89 | 44.70 $\pm$ 1.01 | **45.42 $\pm$ 0.88** |
| Movielens-1M | HR@10 | 72.70 $\pm$ 0.18 | **73.26 $\pm$ 0.20** | 71.91 $\pm$ 0.41 | 70.67 $\pm$ 0.15 |
| | NDCG@10 | 43.04 $\pm$ 0.19 | **44.36 $\pm$ 0.16** | 44.19 $\pm$ 0.15 | 43.74 $\pm$ 0.36 |
| Lastfm-2K | HR@10 | 81.93 $\pm$ 0.80 | **82.38 $\pm$ 0.92** | 81.93 $\pm$ 0.41 | 82.01 $\pm$ 0.64 |
| | NDCG@10 | 72.47 $\pm$ 0.65 | **73.19 $\pm$ 0.38** | 72.41 $\pm$ 0.88 | 72.63 $\pm$ 0.29 |
| Amazon-Video | HR@10 | 59.96 $\pm$ 0.18 | **60.08 $\pm$ 0.08** | 59.75 $\pm$ 0.15 | 59.60 $\pm$ 0.16 |
| | NDCG@10 | **39.14 $\pm$ 0.07** | 39.12 $\pm$ 0.09 | 39.07 $\pm$ 0.14 | 38.99 $\pm$ 0.11 |

Table 6: Performance of different latent embedding sizes on four datasets. The results are the mean and standard deviation of the five repeated trials. Each number has an order of magnitude of 1e-2.

**Effect of negative sample size** We set the negative sample size from 1 to 10 and observe the effect on model performance. Experimental results are shown in Figure 5. For two MovieLens datasets, model performance improves significantly as the number of negative samples increases. Since these two datasets have more user data than Lastfm-2K and Amazon, they contain richer user preference information to learn. Sampling more negative instances help the model to further identify user preferences. On the other two datasets, 4 negative samples are enough to obtain ideal model performance.

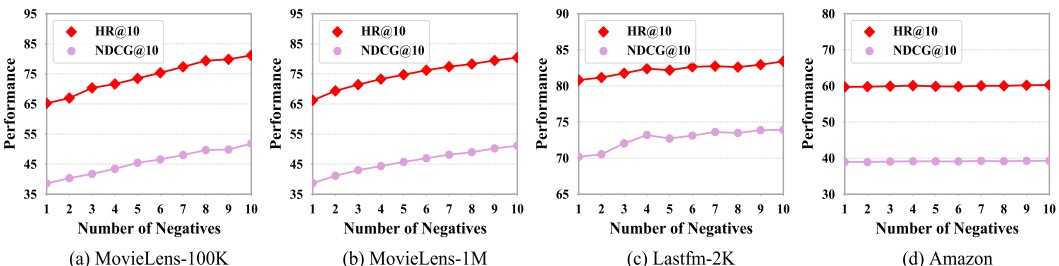

|          |          |          |          |
|----------|----------|----------|----------|
| (a) MovieLens-100K | (b) MovieLens-1M | (c) Lastfm-2K | (d) Amazon |

Figure 5: Performance under different negative sample numbers for each positive sample.

**Effect of client samples participating in each round** There is a trade-off between client sampling ratio and communication efficiency in federated optimization. Generally, the more clients are selected to participate in the global aggregation, the faster the model converges in each round. However, in the physical scenario, it is difficult for the server to collect the complete model information from all the clients. Particularly, there are a large number of user clients in the recommendation scenario, which further increases the difficulty. To verify the relationship between the model's convergence and the clients' participation in each round, we conduct experiments on four datasets with various client samples. To create a consistent validation environment for all datasets, we set the number of users selected in each round as 100, 200, 300, 400 and 500, respectively. Experimental results are represented in Figure 6.

We run the model until convergence and report the best validation performance with the corresponding epoch. According to the experimental results, we can observe that PFedRec could reach consistently advanced performance in all settings on all datasets, even only with 100 clients selected in each round during model training. On the other hand, it is obvious that more clients participating

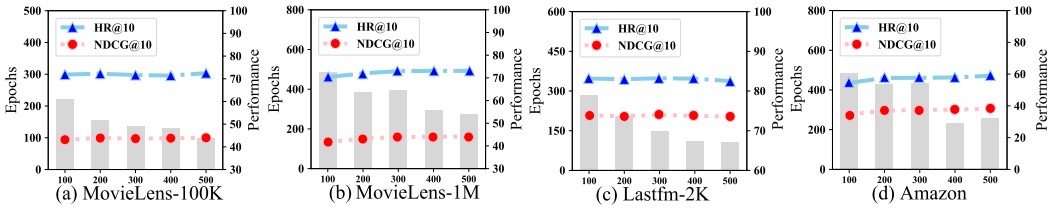

Figure 6: Performance under different client numbers participating in each round.

in each round of training lead to a quicker convergence. PFedRec supports the server to update with insufficient clients accessible, which is ubiquitous in physical circumstances.

### D.4    PROTECTION WITH DIFFERENTIAL PRIVACY

In addition to data locality inherited from the FL framework, introducing privacy-preserving methods into our method can further enhance privacy protection in FedRec. In our model, item embedding is the shared component in the federated optimization process. There is the risk of user interaction information exposure when the client uploads the updated item embedding to the server. Adding noise to the item embedding is a general method to defend against the privacy leakage attack, such as introducing differential privacy and homomorphic encryption into the model. Moreover, designing a reasonable pseudo-interactions injection method is also a potential solution to further enhance privacy protection, which can be discussed in future work.

Here we integrate the local differential privacy technique Choi et al. (2018) into our method as an example. Particularly, we add the zero-mean Laplacian noise to the client's item embedding before uploading to the server,

$$\theta^m = \theta^m + Laplace(0, \lambda) \tag{12}$$

where $\lambda$ is the noise strength. We set $\lambda = [0, 0.1, 0.2, 0.3, 0.4, 0.5]$ to test our method's performance and the results are shown in Table 7. We can see that the performance declines slightly as the noise strength $\lambda$ grows, while the performance drop is still acceptable. For example, when we set $\lambda = 0.4$, the performance is also better than baselines in most cases. Hence, a moderate noise strength is desirable to achieve a good balance between recommendation accuracy and privacy protection.

| Dataset | Noise strength | λ=0 | λ=0.1 | λ=0.2 | λ=0.3 | λ=0.4 | λ=0.5 |
|---|---|---|---|---|---|---|---|
| **ML-100K** | HR@10 | **71.62±0.83** | 71.45±1.01 | 71.26±0.62 | 71.13±1.16 | 70.84±0.96 | 70.88±0.97 |
| | NDCG@10 | **43.44±0.89** | 43.36±0.85 | 43.30±0.81 | 43.22±0.58 | 43.14±0.75 | 43.21±0.69 |
| **ML-1M** | HR@10 | **73.26±0.20** | 73.13±0.11 | 73.19±0.21 | 73.05±0.21 | 73.18±0.29 | 73.08±0.19 |
| | NDCG@10 | **44.36±0.16** | 44.16±0.18 | 44.25±0.32 | 44.26±0.14 | 44.23±0.24 | 44.18±0.40 |
| **Lastfm-2K** | HR@10 | **82.38±0.92** | 82.04±0.63 | 81.91±0.95 | 81.85±0.23 | 81.98±0.52 | 81.88±0.34 |
| | NDCG@10 | **73.19±0.38** | 72.41±0.39 | 72.23±0.49 | 72.43±0.68 | 72.39±0.27 | 72.36±0.42 |
| **Amazon** | HR@10 | **60.08±0.08** | 59.31±0.12 | 59.29±0.04 | 59.21±0.02 | 59.15±0.62 | 59.06±0.71 |
| | NDCG@10 | **39.12±0.09** | 37.97±0.12 | 37.92±0.03 | 37.83±0.05 | 37.81±0.08 | 37.34±0.11 |

Table 7: Results of applying differential privacy technique into our method with various Laplacian noise strength $\lambda$.

### D.5    EVALUATION BASED ON FULL RANKING LIST

As illustrated in C.2, in the main experiments, we adopt the efficient sampled metric. To provide a more comprehensive comparison, we conduct experiments to evaluate the test item in the full ranking list, which is more challenging than the sampled metric. As shown in Table 8, our proposed method still outperforms other baselines in all benchmark datasets, which emphasizes its outstanding efficacy.

| | Method | MovieLens-100K | | MovieLens-1M | | Lastfm-2K | | Amazon-Video | |
|---|---|---|---|---|---|---|---|---|---|
| | | HR@10 | NDCG@10 | HR@10 | NDCG@10 | HR@10 | NDCG@10 | HR@10 | NDCG@10 |
| Rec | NCF | $14.00 \pm 0.62$ | $7.23 \pm 0.53$ | $6.69 \pm 0.39$ | $3.27 \pm 0.37$ | $30.23 \pm 1.18$ | $19.75 \pm 0.34$ | $7.03 \pm 0.17$ | $2.52 \pm 0.35$ |
| | MF | $15.76 \pm 0.68$ | $8.35 \pm 0.43$ | $8.80 \pm 0.19$ | $4.38 \pm 0.14$ | $37.06 \pm 0.43$ | $25.88 \pm 0.50$ | $7.19 \pm 0.23$ | $\mathbf{2.74 \pm 0.31}$ |
| FedRec | FedNCF | $9.95 \pm 0.43$ | $5.03 \pm 0.12$ | $5.70 \pm 0.31$ | $2.87 \pm 0.15$ | $15.43 \pm 1.33$ | $7.09 \pm 1.69$ | $6.96 \pm 0.07$ | $2.44 \pm 0.04$ |
| | FedRecon | $13.10 \pm 0.75$ | $6.92 \pm 0.31$ | $6.80 \pm 0.21$ | $3.25 \pm 0.11$ | $31.63 \pm 0.94$ | $20.00 \pm 2.76$ | $6.70 \pm 0.09$ | $2.05 \pm 0.12$ |
| | FedMF | $16.22 \pm 0.85$ | $8.58 \pm 0.37$ | $8.51 \pm 0.11$ | $4.17 \pm 0.02$ | $38.24 \pm 0.16$ | $19.66 \pm 3.07$ | $7.11 \pm 0.42$ | $2.46 \pm 0.59$ |
| | PFedRec (Ours) | $\mathbf{19.19 \pm 0.32}$ | $\mathbf{10.57 \pm 0.25}$ | $\mathbf{9.75 \pm 0.23}$ | $\mathbf{4.83 \pm 0.15}$ | $\mathbf{56.65 \pm 0.40}$ | $\mathbf{33.11 \pm 5.52}$ | $\mathbf{7.28 \pm 0.12}$ | $2.42 \pm 0.04$ |

Table 8: *Performance of HR@10 and NDCG@10 on full ranking list.*

## D.6 MODEL PERFORMANCE ON INACTIVE USERS

In the recommendation task, there are usually some users with fewer available interactions, and we call them inactive users, which poses a great challenge for model training. Here we show the model performance on inactive users to verify the effectiveness of our method. Particularly, we count the number of ratings from all users, then extract inactive users and calculate their performance. The inactive users' statistics of each dataset are summarized in Table 9. We omit the Amazon-Video due to its high sparsity, where the average size of historical interactions for each user is 8.

| Dataset | Interactions range | Inactive users volume | Ratio of inactive users |
|---|---|---|---|
| ML-100K | 20~30 | 199 | 21.10% |
| ML-1M | 20~30 | 751 | 12.43% |
| Lastfm-2k | 5~20 | 641 | 40.06% |

Table 9: Inactive users' statistics of each dataset.

The comparison results of performance on inactive and full users are shown in Table 10. Our method achieves similar or even better performance on the inactive users set than the full users set, indicating that our method can effectively tackle inactive users.

| User group | MovieLens-100K | | MovieLens-1M | | Lastfm-2K | |
|---|---|---|---|---|---|---|
| | HR@10 | NDCG@10 | HR@10 | NDCG@10 | HR@10 | NDCG@10 |
| Full users | 71.79 | 43.39 | 73.10 | 44.24 | **82.50** | **73.27** |
| Inactive users | **79.40** | **52.51** | **77.76** | **53.24** | 78.94 | 69.08 |

Table 10: *Comparison results of HR@10 and NDCG@10 on inactive users and full users.*

