# OpenReview forum: "Dual personalization for federated recommendation on devices"
_ICLR.cc/2023/Conference — Submitted to ICLR 2023_

### Official Review · Reviewer_rUEH · 2022-10-18

**Confidence:** 3
**Correctness:** 3
**Technical Novelty And Significance:** 3
**Empirical Novelty And Significance:** 3
**Recommendation:** 6

**Clarity, Quality, Novelty And Reproducibility:**

This work is well conducted with good clarity. The contribution of this work is moderate but focused.

**Strength And Weaknesses:**

Strengths:
- This paper is clearly written and easy to follow.
- The proposed method is very simple and easy to use.
- The performance improvement is quite impressive.

Weaknesses:
- It seems that the devices need to maintain a large set of item embeddings, which may be somewhat unscalable. Some discussions on how to reduce the number of locally maintained item embeddings are recommended.
- The authors seem to use sampled metrics over the top recommendation results. In fact, it may not be consistent with the results on the full ranking list (see [1]).
- It is not clear whether the proposed method still works well for inactive users. Since the scoring module may contain much more parameters than the user embedding, it would require a larger number of samples to train.

[1] Krichene, W., & Rendle, S. (2022). On sampled metrics for item recommendation. Communications of the ACM, 65(7), 75-83.

**Summary Of The Paper:**

This paper introduces a dual personalization strategy for federated recommendation, which can better handle non-iid user data and improve recommendation performance. The authors propose to use personalized scoring modules and personalized item embeddings on devices, where the model first updates the scoring module and then updates the item embeddings. Although the proposed method is simple, it is surprisingly effective. The authors also provide some discussions on its insight.

**Summary Of The Review:**

This paper presents a good empirical study on federated recommendation. The proposed method is quite simple, general, and effective. Thus, my recommendation is weak accept, though its technical novelty is not very big.

---

> ### Author Response · Authors · 2022-11-15
> **Response to Reviewer rUEH (1/2)**
>
> We appreciate your appreciation about the valuable contributions of our paper! We have **conducted experiments on the full ranking list** and **presented performance on inactive users**. We try to address your concerns as below.
>
> ### Question 1: [Strength And Weaknesses] Some discussions on how to reduce the number of locally maintained item embeddings are recommended.
> **Response to Q1:** Thanks for your constructive comment. It is not required to download and store the full item embedding table for each device in our method. By sampling 4 negative samples for each positive sample (*Section 6.2 – Baselines and Implementation details*), each client only needs to process 5 times of its interacted items which is much less than the full item table. Besides, in this paper, we mainly focus on the fundamental recommendation task that only relies on the interaction matrix, and it is necessary to instantiate the item embedding module with the item embedding table. When implementing our framework under the scenario where item attributes are available, we can replace the item embedding table with a lightweight neural network similar to the score function. Then the local model on each device can be more lightweight. In the rebuttal revision, we add a new discussion to clarify the communication efficiency in terms of the item embedding table; details can be found in *Appendix B.1 Communication Efficiency (lines 516-529)*.
>
> ### Question 2: [Strength And Weaknesses] This paper uses sampled metrics over the top recommendation results. In fact, it may not be consistent with the results on the full ranking list.
> **Response to Q2:** Thanks for your comment about the evaluation metric. Our experiment follows the common strategy [1, 2] that randomly samples 99 unobserved items for each user and ranks the test item among the 100 items. Besides, we add new experiments according to your recommended paper [3], where we evaluate the test item in the full ranking list, which is more challenging than the previous evaluation setting. As shown in the results below, our proposed method still outperforms other baselines in all benchmark datasets, which emphasizes its outstanding efficacy. In the rebuttal version, we add a subsection to show the empirical results, and details can be found in *Appendix D.5 Evaluation based on Full Ranking List (lines 632-637)*.
> | Method | MovieLens-100K | MovieLens-1M | Lastfm-2K| Amazon-Video|
> |:--------:|:--------:|:--------:|:--------:|:--------:|
> |     | (HR, NDCG)  |(HR, NDCG)|(HR, NDCG)|(HR, NDCG)|
> |NCF|(14.00±0.62, 7.23±0.53)|(6.69±0.39, 3.27±0.37)|(30.23±1.18,	19.75±0.34)|(7.03±0.17,	2.52±0.35)|
> |MF|(15.76±0.68, 8.35±0.43)|(8.80±0.19, 4.38±0.14)|(37.06±0.43, 25.88±0.50)|(7.19±0.23, **2.74±0.31**)|
> |FedNCF|(9.95±0.43, 5.03±0.12)|(5.70±0.31, 2.87±0.15)|(15.43±1.33, 7.09±1.69)|(6.96±0.07, 2.44±0.04)|
> |FedRecon|(13.10±0.75,	6.92±0.31)|(6.80±0.21,	3.25±0.11)|(31.63±0.94,	20.00±2.76)|(6.70±0.09,	2.05±0.12)|
> | FedMF |(16.22±0.85,	8.58±0.37)|(8.51±0.11,	4.17±0.02)|(38.24±0.16,	19.66±3.07)|(7.11±0.42,	2.46±0.59)|
> |  Ours   | (**19.19±0.32**, **10.57±0.25**)|(**9.75±0.23**, **4.83±0.15**)|(**56.65±0.40**,	**33.11±5.52**)|(**7.28±0.12**,	2.42±0.04)|
>
> [1] He, Xiangnan, et al. "Neural collaborative filtering." Proceedings of the 26th international conference on world wide web. 2017.
>
> [2] Koren, Yehuda. "Factorization meets the neighborhood: a multifaceted collaborative filtering model." Proceedings of the 14th ACM SIGKDD international conference on Knowledge discovery and data mining. 2008.
>
> [3] Krichene, W., & Rendle, S. (2022). On sampled metrics for item recommendation. Communications of the ACM, 65(7), 75-83.

---

> ### Author Response · Authors · 2022-11-15
> **Response to Reviewer rUEH (2/2)**
>
> ### Question 3: [Strength And Weaknesses] It is not clear whether the proposed method still works well for inactive users. Since the scoring module may contain much more parameters than the user embedding, it would require a larger number of samples to train.
> **Response to Q3:** Thanks for your comments. Due to the design of dual personalization, the user’s critical information is decomposed into personalized item embeddings and a personalized scoring function. Therefore, in most scenarios, including inactive users, the scoring module does not demand a large neural network. The simple and swift multi-layer neural network is capable of tackling most scenarios. Moreover, to further improve the learning efficiency, we could pre-train a scoring neural model for each user group considering their demographic features.
>
> Empirical results show that our method can effectively tackle inactive users. As illustrated in *Section 6.1- Experimental setup*, each user in the MovieLens dataset has at least 20 rations, and each user in Lastfm-2K and Amazon-Video datasets has at least 5 interactions. Specifically, we add analysis results to demonstrate the effectiveness with inactive users. For each dataset, we count the number of ratings from all users, then extract inactive users and calculate their performance. Since the Amazon-Video is a very sparse dataset, and the average size of historical interactions for each user is 8, resulting in an inactive dataset. Hence, we omit the Amazon-Video dataset and analyze the other three datasets. We summarize the number of inactive users in each dataset and the corresponding performance in the below tables. The comparison results show that our method achieves similar or even better performance on the inactive users set than the full users set, which supports our claim. In the rebuttal version, we add a subsection to show the empirical results, and details can be found in *Appendix D.6 Model Performance on Inactive Users (lines 638-647)*.
> | Dataset | Interactions range | Inactive users volume | Ratio of inactive users |
> |:--------:|:--------:|:--------:|:--------:|
> | MovieLens-100K | 20~30 | 199 | 21.10% |
> | MovieLens-1M | 20~30 | 751 | 12.43% |
> | Lastfm-2K | 5~20 | 641 | 40.06% |
>
> | Method | MovieLens-100K | MovieLens-1M | Lastfm-2K|
> |:--------:|:--------:|:--------:|:--------:|
> |     | (HR, NDCG)  |(HR, NDCG)|(HR, NDCG)|
> | Full users | (71.19, 43.39)|	(73.10,	44.24)|(**82.50**, **73.27**) |
> |  Inactive users   | (**79.40**,	**52.51**)|	(**77.76**,	**53.24**)|	(78.94,	69.08)|
>
> Thanks again for your helpful review and please let us know if there are further questions!

---

> ### Author Response · Authors · 2022-11-27
> **A kind reminder to Reviewer rUEH**
>
> Dear Reviewer rUEH,
>
> We sincerely appreciate your time and effort in reviewing our paper! A big thanks for your positive comments on our work's contributions and our empirical study. We have responded in detail to all your suggestions and concerns in the response and revised paper. Particularly,
> 1. We added a discussion about reducing the number of locally maintained item embeddings (*Appendix B.1, lines 516-527*).
> 2. We conducted new experiments based on the full ranking evaluation setting (*Appendix D.5*).
> 3. We discussed the model's ability to deal with inactive users and added empirical studies to verify the effectiveness (*Appendix D.6*).
>
> Hence, it would be highly appreciated if you could provide feedback about our responses. If you have any further concerns, questions, or suggestions, we are willing to discuss them anytime and include them in the next revision. Thank you!

---

> ### Author Response · Authors · 2022-12-09
> **Discussion stage is about to end**
>
> Dear Reviewer rUEH,
>
> Again, we sincerely thank you for your constructive suggestions that helped to improve our paper! As the deadline for discussion approaches, we sincerely hope to use this opportunity to see if our responses are sufficient and if any concerns remain. It will be our pleasure if you consider updating your review or score. Thanks again for your time and effort!

---

### Official Review · Reviewer_kuz5 · 2022-10-21

**Confidence:** 5
**Correctness:** 2
**Technical Novelty And Significance:** 1
**Empirical Novelty And Significance:** 2
**Recommendation:** 3

**Clarity, Quality, Novelty And Reproducibility:**

a)	Clarity: this paper is well-written and easy to follow.
b)	Quality: the quality of this paper is quite fair due to missing some important baselines.
c)	Novelty: the technical novelty of this paper is very limited
d)	Reproducibility: the authors have provided enough descriptions to reproduce the method.


**Strength And Weaknesses:**

Pros:
a)	The motivation of this paper, i.e., learning a personalized recommendation model for each user, is important.
b)	Empirical results demonstrate the effectiveness of the proposed method.
c)	The paper is well-written and easy to follow.

Cons:
a)  This paper argues that the existing federated recommendation models are cumbersome, so it proposes to learn a LIGHTWEIGHT local score function to make personalized predictions on the client side. However, under the recommendation scenario, the item embedding table generally dominates the vast majority of parameters of the model instead of the parameters of the score function. Thus, the proposed method is meaningless.
b)  The training of the personalized score function and the fine-tuning of the item embedding table are performed locally on the device. However, the training data on each device is very limited, which makes it hard to learn an accurate model.
c)  The technical novelty of this paper is very limited. Learning the personalized item embedding table and score functions has been studied in [1]. The authors should clarify the key difference between their work and [1].
d)  Important baselines are missing, such as [1,2]. The authors should compare the proposed method with them.
e)  There is a duplicate phrase, “optimization framework” in the third contribution on page 2.

Reference:
[1] Lin, Yujie, et al. "Meta matrix factorization for federated rating predictions." Proceedings of the 43rd International ACM SIGIR Conference on Research and Development in Information Retrieval. 2020.
[2] Wu, Chuhan, et al. "A federated graph neural network framework for privacy. preserving personalization." Nature Communications 13.1 (2022): 1-10.


**Summary Of The Paper:**

To learn a personalized and lightweight recommendation model for each client, this paper proposes to learn user-specific lightweight models that are deployed on the device side. Furthermore, it introduces a dual personalization mechanism to learn personalized item embedding tables and the score function for each user. Experimental results on four real-world datasets demonstrate the effectiveness of the proposed method.

**Summary Of The Review:**

Due to the limited novelty and missing important related work,  I recommend reject.

---

> ### Author Response · Authors · 2022-11-15
> **Response to Reviewer kuz5 (1/2)**
>
> Thanks for offering valuable and insightful feedback that has helped us improve the paper's clarity! We are glad for your appreciation of the effectiveness and quality of our paper. We have **clarified contents that are not well explained**, **discussed the differences with related works**, and **conducted experiments of new baselines** in the revision, and we provide the responses below.
>
> ### Question1: [Strength And Weaknesses] The score function is lightweight, while the item embedding table generally dominates the vast majority of model parameters.
> **Response to Q1:** Thanks for the insightful comment. Firstly, it is not required to download and store the full item embedding table for each device in our method. By sampling 4 negative samples for each positive sample (*Section 6.2 – Baselines and Implementation details*), each client only needs to process 5 times of its interacted items, which is much less than the full item table. Besides, in this paper, we mainly focus on the fundamental recommendation task that only relies on the interaction matrix, and it is necessary to instantiate the item embedding module with the item embedding table. When implementing our framework under the scenario where item attributes are available, we can replace the item embedding table with a lightweight neural network similar to the score function. Then the local model on each device can be more lightweight. In the rebuttal revision, we add a new discussion to specifically clarify the communication efficiency in terms of the item embedding table, and details can be found in *Appendix B.1 Communication Efficiency (lines 497-529)*.
>
> ### Question 2: [Strength And Weaknesses] The training of the personalized score function and the fine-tuning of the item embedding table are performed locally on the device. Limited training data on device makes it hard to learn an accurate model.
> **Response to Q2:** Our proposed framework can solve this issue from the below perspectives. First, one-step fine-tuning is an effective way to mitigate the overfitting risk of local model training on the device. Second, the federated learning framework can learn general knowledge by aggregating item embeddings from many devices. Then this knowledge will be sent to each device for further training in the federated optimization process. Empirical results in the experiment indicate that the proposed method gets better recommendation performance than baselines, which also supports our claim. It is also worth noting that, in *Section 6.3 - Table 1*, our proposed method outperforms the baseline methods of MF and NCF trained in a centralized dataset. It indicates that our learning framework can effectively capture fine-grained personalization per user and item, which helps capture user preferences.
>
> ### Question 3: [Strength And Weaknesses] Clarify the differences between the proposed method and the related work MetaMF who also learns personalized item embedding table and score functions to demonstrate the technical novelty.
> **Response to Q3:** Thanks for your suggestion to help improve the clarity. A difference and advantage of our proposed method to the MetaMF is that it does not learn user embeddings and implies personalized information in clients’ local score functions. In the MetaMF, personalization is fulfilled on a server, where a set of user embeddings is maintained to generate personalized item embeddings and score functions. However, learning user embeddings on the server would risk privacy leakage as it collects gradients derived from clients’ local data [1]. On the other hand, aggregating diverse local updates into unified user embeddings would neutralize underlying personal information, which hinders personalization performance. In contrast, only global item embeddings are shared in our method, and differential privacy methods are applicable to ensure privacy safety. Besides, personalization is achieved locally by learning a lightweight score function on each client, where less personal information will be lost. Empirical studies in Q4 validate the superior performance of our methods to the MetaMF.
>
> [1]Geiping, Jonas, et al. "Inverting gradients-how easy is it to break privacy in federated learning?." Advances in Neural Information Processing Systems 33 (2020): 16937-16947.

---

> ### Author Response · Authors · 2022-11-15
> **Response to Reviewer kuz5 (2/2)**
>
> ### Question 4: [Strength And Weaknesses] Important baselines are missing, such as MetaMF and FedGNN.
> **Response to Q4:** We add the comparison analysis with MetaMF and FedGNN. Experimental results of MetaMF are shown in the below table, while the results of FedGNN are still pending since its high communication cost, e.g., about 3 hours are needed for a training round. Compared with MetaMF, our method achieves consistently significant performance improvement, which again demonstrates our model's effectiveness. When all new results are obtained, we will supplement the relevant content in the revised paper.
> | Method |     MovieLens-100K     |      MovieLens-1M      | Lastfm-2K              | Amazon-Video           |
> |:--:|:--:|:--:|:--:|:--:|
> |        |       (HR, NDCG)       |       (HR, NDCG)       | (HR, NDCG)             | (HR, NDCG)             |
> | MetaMF |     (66.38, 40.59)     |     (45.61,	25.24)     | (80.88, 64.24)         | (57.51, 37.25)         |
> |  Ours  | (**71.62**,	**43.44**) | (**73.26**,	**44.36**) | (**82.38**,	**73.19**) | (**60.08**,	**39.12**) |
>
> ### Question 5: Typos in the paper.
> **Response to Q5:** Thanks for pointing it out. We carefully checked the paper and fixed typos. We have booked a professional proofreading service to check the entire article thoroughly.
>
> Thank you again for the comprehensive comments and valuable suggestions. We hope that our replies have clarified your concerns, and we would appreciate it if you could consider raising the rating. Please let us know if you have any further questions, and we look forward to your reply and further discussions, thanks!

---

> ### Author Response · Authors · 2022-11-27
> **A kind reminder to Reviewer kuz5**
>
> Dear Reviewer kuz5,
>
> We sincerely appreciate your time and effort in reviewing our paper! In the response and revised paper, we have made every effort to **address your concerns about model communication efficiency, essential differences compared with related work and more FedRec baselines.** Particularly,
> 1. We added a discussion about reducing the communication cost in terms of the big item embedding table (*Appendix B.1, lines 516-527*).
> 2. We discussed the model's ability to learn with limited training data on the device.
> 3. We clarified the essential differences between our method and MetaMF to demonstrate the technical novelty.
> 4. We add new FedRec baselines to enrich the experiment as you recommended (*Partial results are reported in Response Q4*).
>
> Therefore, can we humbly ask you to read our responses and reconsider your rating of our work? If you have any further concerns, questions, or suggestions, we are willing to discuss them anytime and include them in the next revision. Thank you!

---

> ### Comment · Reviewer_kuz5 · 2022-11-27
> **Response to rebuttal**
>
> Thanks for your response, especially the additional experiments about missing baselines. The authors have addressed part of my concerns. However, I still think that the main technical contribution of learning a lightweight score function is limited, and it is unclear how this approach can be applied in the scenario containing the item embedding table. Therefore, I still stand by my original scores.

---

> > ### Author Response · Authors · 2022-12-09
> > **Further discussion with Reviewer kuz5**
> >
> > Thanks for your kind reply! We are glad to see that our response has addressed part of your concerns. We fully agree with your recognization of the paper's Strength, "The motivation of this paper, i.e., learning a personalized recommendation model for each user, is important.". This paper designs a fundamental personalization mechanism for FedRec, which advances personalization modeling in FedRec.
> >
> > Regarding the concern, "it is unclear how this approach can be applied in the scenario containing the item embedding table." Our method is a distributed learning framework in which each device only needs to update embedding for a small portion of items instead of all items since each user only trains her local model with historical interactions and randomly sampled negative instances, e.g., 5 times of historical interactions. Take Lastfm-2K as an example, the size of the item set is 12,454, and the average size of historical interactions is 116. Each user only needs to update 580 item embeddings on average, accounting for only 4.66% of the entire embedding table, which is further less than transmitting the full item set. Therefore, our method is highly efficient in both communication and computation.
> >
> > To further eliminate potential concerns, we justify our technical contributions from three perspectives.
> >
> > **[Novelty]**
> > 1. Our dual personalization is a **simple-but-effective and novel** mechanism that can significantly outperform all baselines on benchmark datasets. All reviewers agree on this advantage in their comments. Moreover, we have reported all experiments requested by reviewers and the results are all positive.
> > 2. We propose a **fundamental framework** to implement federated recommendations on devices and formulate it as a federated optimization problem.
> >
> > **[Impact]**
> > 1. The proposed framework is general and can be extended to address many other challenges in real applications. Moreover, it provides a new efficient implementation of the federated recommendation. We believe that it will have a fundamental impact on the research community.
> > 2. We provided source codes and additional analysis to enable the research community to reproduce our results.
> >
> > **[Advancement of technology]**
> > 1. Our paper has conducted **a comprehensive analysis** and discussion of the proposed method. For example, we formulated it as a federated optimization framework. The method's privacy-preserving and communication efficiency are discussed and evaluated as well.
> > 2. Rethinking the fundamental architecture of federated recommendation is non-trivial and critical to the community. Most existing works modify an existing recommendation algorithm, such as matrix factorization for a federated learning setting. We instead design a new framework that not only inherits the advantages of the existing recommendation algorithm but also achieves a better learning efficiency in the on-device recommendation setting.
> >
> > Thanks again for your constructive comments! Based on your recognition of the important research motivation of our paper and our detailed response to your concerns, would you mind reconsidering your rating and raising it to reflect the results of our discussion? We sincerely appreciate your support!

---

### Official Review · Reviewer_EAiQ · 2022-10-22

**Confidence:** 3
**Correctness:** 3
**Technical Novelty And Significance:** 2
**Empirical Novelty And Significance:** 1
**Recommendation:** 6

**Clarity, Quality, Novelty And Reproducibility:**

The paper is fairly written but need some polishing to make it more readable.
Novelty is a bit weak as encoding user representation into network parameters could be interpreted as matrix factorization where user embeddings are scoring function parameter (even the model also personalized a bit item embedding with one-step gradient).
The work seems reproduceable since algorithm is clearly given. But, I have no much confidence to reproduce the experiment results.


**Strength And Weaknesses:**

The idea seems straightforward and somewhat novel in terms of reducing modeling cost.
The description of this paper is a bit lack polishing.
I am also concerning the risk of overfitting of this model.




**Summary Of The Paper:**

This paper presents a new personalized federated training model architecture that does not contain user embeddings explicitly. The model relies on personalized item embedding and scoring function to achieve the personalized recommendation. When conducting federated training (global), it learns a shared embedding parameter for the items. And, during the personalization step, it conducts 1 step gradient fine-tuning for each client. The term "Dual" personalization simply comes from separately learned  scoring function that never been federated trained. Experiment results show promising performance improvements.

**Summary Of The Review:**

The idea seems straightforward and somewhat novel in terms of reducing modeling cost. But the cost reduction is not significant (as least not demonstrated to be significant).
The description of this paper is a bit lack polishing. And, I have to guess what the authors try to deliver
- 1. does \theta^m contains item embedding? why do you use E and \theta^m together? are they somewhat different?
- 2. why is one-step gradient for each client sufficient? any justification or assumption?
The scoring functions are never transferred and there is no default scoring function. In this case, how do you handle users who lack interactions? I think this need further discussion.
I am also concerning the risk of overfitting of this model. As the model personalizing (fine-tuning) ALL of the client parameters. The strong collaborative regularization is missing. Considering the datasets used in this paper are often showing very coherent preference (lack diversity) of each user, it is possible that the client model is simply learned to remember the user's historical interactions.

---

> ### Author Response · Authors · 2022-11-15
> **Response to Reviewer EAiQ (1/2)**
>
> We thank the reviewer for the appreciation and valuable comments on our paper and we try to address your concerns as below.
>
> ### Question 1: [Strength And Weakness] The description of this paper is a bit lack polishing.
> **Response to Q1:** Thanks for the comments. We have booked a professional proofreading service to polish the entire article.
>
> ### Question 2: [Strength And Weakness] Concerning the risk of overfitting of this model.
> **Response to Q2:** Thanks for your question. Our proposed algorithm can solve this concern from three perspectives. First, one-step fine-tuning can effectively mitigate the risk of overfitting the personalized model on each device. More details can be found in the below response to Question 6. Second, as replied in Questions 7 & 8, the user-specific score function can also be effectively trained by jointly optimizing with the item embedding, a shared component collecting common information from all users. Moreover, a pre-trained score function will be an easy-to-implement solution to further enhance the performance. The experimental results can support our current solutions can effectively tackle this scenario because the datasets include many users with very few interactions, such as 5 ~ 30.
>
> ### Question 3: [Clarity, Quality, Novelty And Reproducibility] Novelty is a bit weak as encoding user representation into network parameters could be interpreted as matrix factorization where user embeddings are scoring function parameter (even the model also personalized a bit item embedding with one-step gradient).
> **Response to Q3:** Thanks for your high-level summary of the novelty of our method. As we discussed in *Section 5.2 A General Framework for Federated Recommendation*, our proposed neural architecture is naturally aligned to the layer-wise neural network architecture, so it enjoys more learning capacity and is more flexible to be extended with existing techniques. For example, we can equip the score functions of clients with different neural network architectures to provide personalized services based on the client type and local data characteristics. Besides, as illustrated in *Section 6.4 Integrating Baselines with Our Dual Personalization Mechanism*, our method can be easily introduced to existing FedRec to enhance its performance as a convenient plug-and-play component. Imaging the future technique evolution, the models derived from our proposed framework could be significantly distinct from the MF-based models.
>
> ### Question 4: [Clarity, Quality, Novelty And Reproducibility] The work seems reproducible since the algorithm is clearly given. But, I have no much confidence to reproduce the experimental results.
> **Response to Q4:** Thanks for your comments. Kindly remind that all source code and detailed experiment configurations are presented in the *Supplementary Material*. Moreover, we will open the source code in the formal version to enhance reproducibility.
>
> ### Question 5: [Summary Of The Review] Does \theta^m contains item embedding? why do you use E and \theta^m together? are they somewhat different?
> **Response to Q5:** Thank you for suggesting the clarification. $E$ represents the item embedding learning function parameterized by $\theta^m$. We use $E$ as a general form to describe the processing of the item embedding. This paper implements $E$ with an item embedding layer parameterized by $\theta^m$. However, the $E$ could also be implemented in other ways in traditional recommendation algorithms, such as MLP or CNN.

---

> ### Author Response · Authors · 2022-11-15
> **Response to Reviewer EAiQ (2/2)**
>
> ### Question 6: [Summary Of The Review] Why is one-step gradient for each client sufficient? any justification or assumption?
> **Response to Q6:** The one-step gradient fine-tuning is necessary for our proposed PFedRec, and we would like to refer to the theoretical analysis in [1] to support our design. A significant consideration is that the PFedRec alternately tunes one gradient step each for the global item embedding module and the client-specific score function. This learning process is consistent with the partial personalization scheme studied in [1], where theoretical analysis shows that one gradient step is sufficient and the algorithm’s convergence is guaranteed when the following assumptions hold:
>   1) the gradient of a local loss function is L-Lipschitz smooth,
>   2) the gradient of a local loss function has bounded variance, and
>   3) the variance of the gradient of local loss functions on different clients is bounded.
>
> As the three assumptions are universal and hold for most functions, the conclusions from [1] also hold in our proposed PFedRec, i.e., the one-step fine-tuning is sufficient and guaranteed to converge. We add discussion in *Appendix A.2 Effectiveness of One-Step Fine-Tuning (lines 484-495)* in the revision.
>
> [1] Pillutla, Krishna, et al. "Federated learning with partial model personalization." International Conference on Machine Learning. PMLR, 2022.
>
> ### Question 7: [Summary Of The Review] The scoring functions are never transferred and there is no default scoring function. In this case, how do you handle users who lack interactions?
> **Response to Q7:** Thanks for the comments. We believe this scenario could be identified as a cold-start problem of recommendation. If a new user has no or very few interactions, the recommendation system generally recommends the most popular items to the user, or recommends the items based on similar users by considering their demographic features. If the user has several interactions only, then an appropriate strategy could be recommending similar items of the interacted items.
>
> Our proposed framework can be applied to solve the cold-start problem by integrating it with existing recommendation techniques. First, in our framework, item embedding is a shared component aggregated globally with the locally updated item embedding from different users. Users with similar preferences collaboratively obtain the common item embedding describing the correlations between items. Training the client model with the enhanced item embedding introduces the auxiliary information from similar users, hence improving the recommendation performance of users who lack interactions. Second, to tackle the cold start problem of new users, we can pre-train a score function to learn the popular preference of a population group according to their demographic features, and then apply this global model as an initialization.
>
> It is worth noting, as we mentioned in the *Conclusion (Section 7)*, “Given the complex nature of modern recommendation applications, such as cold-start problems, dynamics, using auxiliary information, and processing multi-modality contents, our proposed framework is simple and flexible enough to be extended to tackle many new challenges.”
>
> ### Question 8: [Summary Of The Review] The model has the risk of overfitting since the model personalizing all of the client parameters with missing collaborative regularization and it is possible that the client model is simply learned to remember the user's historical interactions.
> **Response to Q8:** As we respond in Q6, one-step fine-tuning is an effective way to mitigate the risk of overfitting. In each round, the client performs local gradient steps based on the common item embedding from the server. Once the local item embeddings are updated, they will be uploaded to the server for a new round of collaborative aggregation. The item embedding is transmitted between the server and clients iteratively during federated optimization. Besides, for dataset split, we follow the prevalent leave-one-out evaluation and report the recommendation performance on the user’s latest interaction, which is invisible during training. The experimental results can support our claims.
>
> Thanks for your encouraging feedback again! We hope our responses have clarified your concerns and please let us know if you have any further questions.

---

> ### Author Response · Authors · 2022-11-27
> **A kind reminder to Reviewer EAiQ**
>
> Dear Reviewer EAiQ,
>
> We sincerely appreciate your time and effort in reviewing our paper! We believe your constructive comments will further strengthen our paper, especially the clarification about model capability. In the response and revised paper, we have made every effort to address your concerns. Particularly,
> 1. We discussed the potential generality and flexibility compared with related work to clarify the contributions.
> 2. We added a discussion about the reason for the one-step fine-tuning design in our method (*Appendix A.2*).
> 3. We discussed the ability of our model to inactive users and mitigating the risk of overfitting.
> 4. We clarified the differences between two related notations and added experimental details (*Appendix C.1 and C.3*) to facilitate reproducibility.
>
> Hence, it would be highly appreciated if you could provide feedback on our responses or confirm whether there is no remained concern. If you have any further concerns, questions, or suggestions, we are willing to discuss and reflect on them in the next revision. Thank you!

---

> ### Author Response · Authors · 2022-12-09
> **Discussion stage is about to end**
>
> Dear Reviewer EAiQ,
>
> Again, we sincerely thank you for your constructive suggestions that helped to improve our paper! As the deadline for discussion approaches, we sincerely hope to use this opportunity to see if our responses are sufficient and if any concerns remain. It will be our pleasure if you consider updating your review or score. Thanks again for your time and effort!

---

### Official Review · Reviewer_2m4p · 2022-10-27

**Confidence:** 4
**Correctness:** 4
**Technical Novelty And Significance:** 3
**Empirical Novelty And Significance:** 3
**Recommendation:** 5

**Clarity, Quality, Novelty And Reproducibility:**

The paper is well presented, which is easy to be understood.

It is not difficult to reproduce the results (though not very easy). The authors are courageed to include more details about empirical studies, e.g., parameter configurations and data preprocessing.


**Strength And Weaknesses:**

Strength:
1 The idea of using personalized item embedding in federated recommendation is new (though it is not new in traditional recommendation).

Weakness:
1 For key issues in federated recommendation, the authors do not contribute/discuss much, e.g., communication cost, privacy protection, time complexity.

2 The technical contribution is limited. For example, the contents of Section 4 are not about a formal and principled solution, but most about heuristics.

3 For the studied problem, there are many recent works, which are not studied in the experiments.

**Summary Of The Paper:**

The authors study the problem of federated recommendation with implicit feedback, and propose a new framework with personalized score function and personalized item embedding (without user embedding), which is illustrated in Figure 1(c).

**Summary Of The Review:**

The authors study an important problem of federated recommendation with implicit feedback, and propose a new framework with personalized score function and personalized item embedding.

My major concern is that the technical contribution compared with existing works on federated recommendation is limited.

Minors:
optimization framework optimization framework
It is a federated version of FedNCF

---

> ### Author Response · Authors · 2022-11-15
> **Response to Reviewer 2m4p (1/2)**
>
> We appreciate your constructive feedback and valuable suggestions for our work! We have **added discussions about key issues in FedRec**, **supplemented empirical study details**, and **fixed the typos**. We present our responses as follows.
>
> ### Question 1: [Strength And Weaknesses] More discussion about key issues in federated recommendation, e.g., communication cost, privacy protection and time complexity.
> **Response to Q1:** Thanks for your helpful suggestions. We add discussions about the above key issues in FedRec in the appendix. Our proposed model has an edge on communication efficiency and achieves steady performance under privacy enhancement circumstances.
>
> Particularly, we analyze the factors that determine the communication cost and time complexity and formulate in *Appendix B.1 Communication Efficiency (lines 497-529)* and *Appendix B.2 Time Complexity (lines 530-536)*, respectively. For privacy protection, we discuss the potential privacy enhancement method of our model and integrate the Local Differential Privacy technique into our method as an example. Also, empirical experiments are conducted to analyze model performance under privacy protection, and details are shown in *Appendix D.4 Protection with Differential Privacy (lines 615-631)*. Please kindly refer to the corresponding parts in the revised manuscript for details.
>
> ### Question2: [Strength And Weaknesses] Limited technical contribution. For example, the contents of Section 4 are not about a formal and principled solution, but most about heuristics.
> **Response to Q2:** Thanks for your constructive comments. We kindly explain our formulation and complete pseudo-code introducing our method. Particularly, we devise a bi-level optimization objective (Eq.6, Eq.7), where a "good" global item embedding initialization is learned for all clients and clients can quickly adapt the device learning task based on the local data (Eq.5, Eq.8). Furthermore, to solve the optimization problem, we conduct an alternative optimization algorithm to train the model (Algorithm 1). To demonstrate our method more clearly, we refine the corresponding contents in *Section 4.1 Objective Function* and add discussion about the proposed dual personalization mechanism in *Appendix A More Discussion about Dual Personalization (lines 473-495)* in the revision.
>
> Besides, we discuss the differences between our method and existing works to clarify the technical contribution. Our key insight is to model user personalization in the FedRec framework. To this end, we propose a dual personalization to learn personalized score function and item embedding. In traditional recommendation frameworks, the score function is usually a common component and is shared among users. Some works propose to model personalized item embedding by virtue of the user’s historical context interactions, while the item embedding is also shared among users. In our method, we learn user-specific item embedding based on the user’s personal interaction data, which is fine-grained personalization. Compared with existing FedRec methods, which usually model user personalization with user embedding and leave the item embedding as the common role, our method goes one step further and learns personalized item embedding for each user. Particularly, we first learn an average global model and update once with local data, which is a simple and general mechanism that can be easily integrated into existing methods to enhance them. Besides, we replace user embedding with the flexible score function, which is a general form of federated recommendation.

---

> ### Author Response · Authors · 2022-11-15
> **Response to Reviewer 2m4p (2/2)**
>
> ### Question 3: [Strength And Weaknesses] Missing recent works in experiment.
> **Response to Q3:** Thanks for the question about baselines which need to be better discussed. In this paper, we present a fundamental operation, namely the dual personalization mechanism, and then verify its efficacy and compatibility with existing methods. Focusing on the performance improvement of the infrastructure of recommendation models that all others derive from, we select the general and fundamental baselines that conduct recommendations based on the interaction matrix.
>
> For centralized recommendation models, we select the most representative NCF [1] and MF [2]. Federated recommendation is a new developing research direction with limited work. We surveyed the closely related FedRec methods thoroughly and selected the pioneering work FedMF [3], the recent model FedNCF [4] and the advanced personalized FL method FedRecon [5] as the federated baselines. We omitted works relying on more data sources in modeling that are orthogonal with our mechanism, such as FedFast [6] and Efficient-FedRec [7]. Besides, methods omitted due to the risk of privacy leakages, such as MetaMF [8] and FedGNN [9], whose server preserves all the recommendation model parameters. We have improved the organization of *Related Work (Section 2.2 Federated Recommendation Systems)* to present a more carefully designed taxonomy of federated recommendation. Please kindly find the updates in the revision.
>
> To further enrich the FedRec baselines, we take MetaMF and FedGNN as new baselines into our experiments. Experimental results of MetaMF are summarized in the table below, while the results of FedGNN are still pending due to its high communication cost, e.g., about 3 hours for one training round. Our model still takes the leading position, which validates its efficacy against state-of-the-art methods. When all new results are obtained, we will supplement the relevant content in the revised paper.
> | Method | MovieLens-100K | MovieLens-1M | Lastfm-2K| Amazon-Video|
> |:--------:|:--------:|:--------:|:--------:|:--------:|
> |     | (HR, NDCG)  |(HR, NDCG)|(HR, NDCG)|(HR, NDCG)|
> | MetaMF | (66.38, 40.59)|	(45.61,	25.24)|(80.88, 64.24)	|(57.51, 37.25)  |
> |  Ours   | (**71.62**,	**43.44**)|	(**73.26**,	**44.36**)|	(**82.38**,	**73.19**)|	(**60.08**,	**39.12**)|
>
>
> ### Question 4: [Clarity, Quality, Novelty And Reproducibility] More details about empirical studies, e.g., parameter configurations and data preprocessing.
> **Response to Q4:** Thanks for pointing it out. We add the parameter configurations in *Appendix C.3 Parameter Configuration (lines 556-562)* and data preprocessing descriptions in *Appendix C.1 More Details of the Datasets (lines 538-542)* in the revision. Moreover, the source codes are provided in the supplementary materials. We will also open-source our code soon to enhance the reproducibility of this work.
>
> ### Question 5: [Summary Of The Review] Typos in the paper.
> **Response to Q5:** Thanks for pointing it out. We carefully checked the paper and fixed typos. We have booked a professional proofreading service to check the entire article thoroughly.
>
> Thank you again for your helpful and thoughtful comments! We hope our responses address your concern. In light of these clarifications, we would appreciate it if you could consider increasing your rating and confidence score. If there are any further questions, we are very glad to continue the discussion!

---

> ### Author Response · Authors · 2022-11-27
> **A kind reminder to Reviewer 2m4p**
>
> Dear Reviewer 2m4p,
>
> We sincerely appreciate your time and effort in reviewing our paper! We believe that your constructive comments will further strengthen our paper. In the response and revised paper, we have made every effort to **address your concerns about key issues discussion in FedRec, technique contribution compared with existing FedRec, more FedRec baselines in the experiment and more details about empirical studies**. Particularly,
> 1. We added discussions about communication cost (*Appendix B.1*), time complexity (*Appendix B.2*) and privacy protection (*Appendix D.4*).
> 2. We refined the Section method and added discussion about the proposed dual personalization mechanism (*Appendix A*) to clary the technique contribution.
> 3. We reorganized Section related work and added new FedRec baselines in the experiments (*Partial results are reported in Response Q3*).
> 4. We added more experimental details (*Appendix C.1 and C.3*) to accelerate the result's reproducibility.
>
> Therefore, can we humbly ask you to read our responses and reconsider your rating of our work? If you have any further concerns, questions, or suggestions, we are willing to discuss them anytime and include them in the next revision. Thank you!

---

> ### Author Response · Authors · 2022-12-09
> **Discussion stage is about to end**
>
> Dear Reviewer 2m4p,
>
> Again, we sincerely thank you for your constructive suggestions that helped to improve our paper! As the deadline for discussion approaches, we sincerely hope to use this opportunity to see if our responses are sufficient and if any concerns remain. It will be our pleasure if you consider updating your review or score. Thanks again for your time and effort!

---

### Author Response · Authors · 2022-11-17
**Paper Revision Summary**

## Paper Revision Summary
We would like to express our sincere gratitude to all reviewers for their constructive comments. We carefully refined the paper according to the comments and introduced several changes, which are highlighted in blue color in the latest PDF (main paper and appendix) and briefly summarized below.

1. [Reorganize Related Work] We improved Section 2.2 (Related work - federated recommendation system) by categorizing all recent work into two classes.
2. [Clarified Description] To better present our technical contribution in principle, we improved the wording in Section 4.1 (Objective function), and added more discussion about Section 4.2 (Dual Personalization) in Appendix A.
3. [More Analysis] We added more analysis of our method in Appendix B as an extension of Section 5 (Discussion). Specifically, we evaluated the communication efficiency and computation time complexity. We further added a discussion of privacy and an experiment in Appendix D.4.
4. [Better Reproducibility] To improve the reproducibility, we provide the detailed hyperparameter configuration in Appendix C.3. By using this configuration in our submitted source code, readers can easily reproduce the results in our paper.
5. [More Experiments] We added more experiments to further strengthen the empirical analysis, including (1) evaluation on the full ranking list, (2) FedRec+Differential Privacy, and (3) performance analysis on inactive users. Due to the limited time for rebuttal, we are still running experiments for some baselines and provided part of the comparison result in the response to the reviewers for now. We will share new updates with reviewers once getting the complete experimental results.

---

### Author Response · Authors · 2022-11-27
**Gentle Reminder for Discussion (No response yet)**

Dear AC and Reviewers,

We sincerely appreciate the valuable comments from all reviewers, which are very helpful in improving the paper. We have carefully addressed the reviewer's concerns one by one in the response with corresponding clarifications and additional experiments added to the revision. We are more than happy to address any further concerns and questions from the reviewers during this discussion phase. **However, up to now we have not heard anything from any reviewer yet.** We would greatly appreciate your help if you could remind the reviewers to join the discussion. Thank you very much for the time and effort you put on our submission!

---

### Decision · Program_Chairs · 2023-01-20

**Decision:**

Reject

**Justification For Why Not Higher Score:**

The paper has a few flaws that does not merit a higher score

**Justification For Why Not Lower Score:**

N/A

**Metareview: Summary, Strengths And Weaknesses:**

In this paper, the authors propose a Personalized Federated Recommendation (PFedRec) framework for providing privacy-preserving recommendation services in federated settings. The framework learns many user-specific lightweight models that can be deployed on smart devices, rather than a single heavyweight model on a server. A dual personalization mechanism is also proposed to effectively learn fine-grained personalization on both users and items. The overall learning process is formulated as a unified federated optimization framework. Experiments on multiple benchmark datasets demonstrate the effectiveness of PFedRec and the dual personalization mechanism. Visualizations and analysis of the personalization techniques in item embedding are also provided. The reviewers thought that learning a personalized recommendation model for each user is interesting and important, the paper is easy to follow and  that the paper has nice empirical results demonstrate the effectiveness of the proposed method. The reviewers raised concerns about various claims including that "the item embedding table generally dominates the vast majority of parameters of the model instead of the parameters of the score function", the use of on device learning, lack of substantial technical novelty, missing baseline comparisons. Finally, one reviewer raised the concern that "The authors seem to use sampled metrics over the top recommendation results. In fact, it may not be consistent with the results on the full ranking list (see [1]).". The authors provided a comprehensive response. However, IMO the authors response does not properly address the issues that were raised. In its current form the paper requires substantial revision and therefore I can not recommend acceptance. That said I think is an interesting paper with potential and recommend resubmission to a future venue following the suggestion above.